# Evaluating the Robustness of Neural Networks: An Extreme Value Theory Approach

**Tsui-Wei Weng[1][*], Huan Zhang[2][*], Pin-Yu Chen[3], Jinfeng Yi[4], Dong Su[3], Yupeng Gao[3], Cho-Jui Hsieh[2], Luca Daniel[1]**

[1]Massachusetts Institute of Technology, Cambridge, MA 02139
[2]University of California, Davis, CA 95616
[3]IBM Research AI, Yorktown Heights, NY 10598
[4]Tencent AI Lab, Bellevue, WA 98004
`twweng@mit.edu, ecezhang@ucdavis.edu,`
`pin-yu.chen@ibm.com, jinfengyi.ustc@gmail.com,`
`{dong.su,yupeng.gao}@ibm.com, chohsieh@ucdavis.edu, dluca@mit.edu`

## Abstract

The robustness of neural networks to adversarial examples has received great attention due to security implications. Despite various attack approaches to crafting visually imperceptible adversarial examples, little has been developed towards a comprehensive measure of robustness. In this paper, we provide a theoretical justification for converting robustness analysis into a local Lipschitz constant estimation problem, and propose to use the Extreme Value Theory for efficient evaluation. Our analysis yields a novel robustness metric called CLEVER, which is short for **C**ross **L**ipschitz **E**xtreme **V**alue for n**E**twork **R**obustness. The proposed CLEVER score is attack-agnostic and computationally feasible for large neural networks. Experimental results on various networks, including ResNet, Inception-v3 and MobileNet, show that (i) CLEVER is aligned with the robustness indication measured by the $\ell_2$ and $\ell_\infty$ norms of adversarial examples from powerful attacks, and (ii) defended networks using defensive distillation or bounded ReLU indeed achieve better CLEVER scores. To the best of our knowledge, CLEVER is the first attack-independent robustness metric that can be applied to any neural network classifier.

## 1 Introduction

Recent studies have highlighted the lack of robustness in state-of-the-art neural network models, e.g., a visually imperceptible adversarial image can be easily crafted to mislead a well-trained network (Szegedy et al., 2013; Goodfellow et al., 2015; Chen et al., 2017a). Even worse, researchers have identified that these adversarial examples are not only valid in the digital space but also plausible in the physical world (Kurakin et al., 2016a; Evtimov et al., 2017). The vulnerability to adversarial examples calls into question safety-critical applications and services deployed by neural networks, including autonomous driving systems and malware detection protocols, among others.

In the literature, studying adversarial examples of neural networks has twofold purposes: (i) *security implications*: devising effective attack algorithms for crafting adversarial examples, and (ii) *robustness analysis*: evaluating the intrinsic model robustness to adversarial perturbations to normal examples. Although in principle the means of tackling these two problems are expected to be independent, that is, the evaluation of a neural network's intrinsic robustness should be agnostic to attack methods, and vice versa, existing approaches extensively use different attack results as a measure of robustness of a target neural network. Specifically, given a set of normal examples, the attack success rate and distortion of the corresponding adversarial examples crafted from a particular attack algorithm are treated as robustness metrics. Consequently, the network robustness is entangled with the attack algorithms used for evaluation and the analysis is limited by the attack capabilities. More importantly, the dependency between robustness evaluation and attack approaches can cause

---

[*]Tsui-Wei Weng and Huan Zhang contributed equally

biased analysis. For example, *adversarial training* is a commonly used technique for improving the robustness of a neural network, accomplished by generating adversarial examples and retraining the network with corrected labels. However, while such an adversarially trained network is made robust to attacks used to craft adversarial examples for training, it can still be vulnerable to unseen attacks.

Motivated by the evaluation criterion for assessing the quality of text and image generation that is completely independent of the underlying generative processes, such as the BLEU score for texts (Papineni et al., 2002) and the INCEPTION score for images (Salimans et al., 2016), we aim to propose a comprehensive and attack-agnostic robustness metric for neural networks. Stemming from a perturbation analysis of an arbitrary neural network classifier, we derive a universal lower bound on the minimal distortion required to craft an adversarial example from an original one, where the lower bound applies to any attack algorithm and any $\ell_p$ norm for $p \geq 1$. We show that this lower bound associates with the maximum norm of the local gradients with respect to the original example, and therefore robustness evaluation becomes a local Lipschitz constant estimation problem. To efficiently and reliably estimate the local Lipschitz constant, we propose to use *extreme value theory* (De Haan & Ferreira, 2007) for robustness evaluation. In this context, the extreme value corresponds to the local Lipschitz constant of our interest, which can be inferred by a set of independently and identically sampled local gradients.With the aid of extreme value theory, we propose a robustness metric called CLEVER, which is short for **C**ross **L**ipschitz **E**xtreme **V**alue for n**E**twork **R**obustness. We note that CLEVER is an attack-independent robustness metric that applies to any neural network classifier. In contrast, the robustness metric proposed in Hein & Andriushchenko (2017), albeit attack-agnostic, only applies to a neural network classifier with one hidden layer.

We highlight the main contributions of this paper as follows:

- We propose a novel robustness metric called CLEVER, which is short for **C**ross **L**ipschitz **E**xtreme **V**alue for n**E**twork **R**obustness. To the best of our knowledge, CLEVER is the first robustness metric that is attack-independent and can be applied to any arbitrary neural network classifier and scales to large networks for ImageNet.
- The proposed CLEVER score is well supported by our theoretical analysis on formal robustness guarantees and the use of extreme value theory. Our robustness analysis extends the results in Hein & Andriushchenko (2017) from continuously differentiable functions to a special class of non-differentiable functions – neural+ networks with ReLU activations.
- We corroborate the effectiveness of CLEVER by conducting experiments on state-of-the-art models for ImageNet, including ResNet (He et al., 2016), Inception-v3 (Szegedy et al., 2016) and MobileNet (Howard et al., 2017). We also use CLEVER to investigate defended networks against adversarial examples, including the use of defensive distillation (Papernot et al., 2016) and bounded ReLU (Zantedeschi et al., 2017). Experimental results show that our CLEVER score well aligns with the attack-specific robustness indicated by the $\ell_2$ and $\ell_\infty$ distortions of adversarial examples.

## 2 BACKGROUND AND RELATED WORK

### 2.1 ATTACKING NEURAL NETWORKS USING ADVERSARIAL EXAMPLES

One of the most popular formulations found in literature for crafting adversarial examples to mislead a neural network is to formulate it as a minimization problem, where the variable $\boldsymbol{\delta} \in \mathbb{R}^d$ to be optimized refers to the perturbation to the original example, and the objective function takes into account unsuccessful adversarial perturbations as well as a specific norm on $\boldsymbol{\delta}$ for assuring similarity. For instance, the success of adversarial examples can be evaluated by their cross-entropy loss (Szegedy et al., 2013; Goodfellow et al., 2015) or model prediction (Carlini & Wagner, 2017b). The norm constraint on $\boldsymbol{\delta}$ can be implemented in a clipping manner (Kurakin et al., 2016b) or treated as a penalty function (Carlini & Wagner, 2017b). The $\ell_p$ norm of $\boldsymbol{\delta}$, defined as $\|\boldsymbol{\delta}\|_p = (\sum_{i=1}^d |\boldsymbol{\delta}_i|^p)^{1/p}$ for any $p \geq 1$, is often used for crafting adversarial examples. In particular, when $p = \infty$, $\|\boldsymbol{\delta}\|_\infty = \max_{i \in \{1,...,d\}} |\boldsymbol{\delta}_i|$ measures the maximal variation among all dimensions in $\boldsymbol{\delta}$. When $p = 2$, $\|\boldsymbol{\delta}\|_2$ becomes the Euclidean norm of $\boldsymbol{\delta}$. When $p = 1$, $\|\boldsymbol{\delta}\|_1 = \sum_{i=1}^p |\boldsymbol{\delta}_i|$ measures the total variation of $\boldsymbol{\delta}$. The state-of-the-art attack methods for $\ell_\infty$, $\ell_2$ and $\ell_1$ norms are the iterative fast gradient sign method (I-FGSM) (Goodfellow et al., 2015; Kurakin et al., 2016b), Carlini and Wagner's attack (CW attack) (Carlini & Wagner, 2017b), and elastic-net attacks to deep neural networks (EAD) (Chen et al., 2017b), respectively. These attacks fall into the category of *white-box* attacks since the network model is assumed to be transparent to an attacker. Adversarial examples

can also be crafted from a *black-box* network model using an ensemble approach (Liu et al., 2016), training a substitute model (Papernot et al., 2017), or employing zeroth-order optimization based attacks (Chen et al., 2017c).

## 2.2 Existing Defense Methods

Since the discovery of vulnerability to adversarial examples (Szegedy et al., 2013), various defense methods have been proposed to improve the robustness of neural networks. The rationale for defense is to make a neural network more resilient to adversarial perturbations, while ensuring the resulting defended model still attains similar test accuracy as the original undefended network. Papernot et al. proposed *defensive distillation* (Papernot et al., 2016), which uses the distillation technique (Hinton et al., 2015) and a modified softmax function at the final layer to retrain the network parameters with the prediction probabilities (i.e., soft labels) from the original network. Zantedeschi et al. (2017) showed that by changing the ReLU function to a bounded ReLU function, a neural network can be made more resilient. Another popular defense approach is *adversarial training*, which generates and augments adversarial examples with the original training data during the network training stage. On MNIST, the adversarially trained model proposed by Madry et al. (2017) can successfully defend a majority of adversarial examples at the price of increased network capacity. *Model ensemble* has also been discussed to increase the robustness to adversarial examples (Tramèr et al., 2017; Liu et al., 2017). In addition, *detection* methods such as feature squeezing (Xu et al., 2017) and example reforming (Meng & Chen, 2017) can also be used to identify adversarial examples. However, the CW attack is shown to be able to bypass 10 different detection methods (Carlini & Wagner, 2017a). In this paper, we focus on evaluating the intrinsic robustness of a neural network model to adversarial examples. The effect of detection methods is beyond our scope.

## 2.3 Theoretical Robustness Guarantees for Neural Networks

Szegedy et al. (2013) compute global Lipschitz constant for each layer and use their product to explain the robustness issue in neural networks, but the global Lipschitz constant often gives a very loose bound. Hein & Andriushchenko (2017) gave a robustness lower bound using a local Lipschitz continuous condition and derived a closed-form bound for a multi-layer perceptron (MLP) with a single hidden layer and softplus activation. Nevertheless, a closed-form bound is hard to derive for a neural network with more than one hidden layer. Wang et al. (2016) utilized terminologies from topology to study robustness. However, no robustness bounds or estimates were provided for neural networks. On the other hand, works done by Ehlers (2017); Katz et al. (2017a;b); Huang et al. (2017) focus on formally verifying the viability of certain properties in neural networks for any possible input, and transform this formal verification problem into satisfiability modulo theory (SMT) and large-scale linear programming (LP) problems. These SMT or LP based approaches have high computational complexity and are only plausible for very small networks.

Intuitively, we can use the distortion of adversarial examples found by a certain attack algorithm as a robustness metric. For example, Bastani et al. (2016) proposed a linear programming (LP) formulation to find adversarial examples and use the distortions as the robustness metric. They observe that the LP formulation can find adversarial examples with smaller distortions than other gradient-based attacks like L-BFGS (Szegedy et al., 2013). However, the distortion found by these algorithms is an *upper bound* of the true minimum distortion and depends on specific attack algorithms. These methods differ from our proposed robustness measure CLEVER, because CLEVER is an estimation of the *lower bound* of the minimum distortion and is *independent* of attack algorithms. Additionally, unlike LP-based approaches which are impractical for large networks, CLEVER is computationally feasible for large networks like Inception-v3. The concept of minimum distortion and upper/lower bound will be formally defined in Section 3.

## 3 Analysis of Formal Robustness Guarantees for a Classifier

In this section, we provide formal robustness guarantees of a classifier in Theorem 3.2. Our robustness guarantees are general since they only require a mild assumption on Lipschitz continuity of the classification function. For differentiable classification functions, our results are consistent with the main theorem in (Hein & Andriushchenko, 2017) but are obtained by a much simpler and more

Table 1: Table of Notation

| Notation | Definition | Notation | Definition |
|---|---|---|---|
| $d$ | dimensionality of the input vector | $\Delta_{p,\min}$ | minimum $\ell_p$ distortion of $\boldsymbol{x_0}$ |
| $K$ | number of output classes | $\beta_L$ | lower bound of minimum distortion |
| $f : \mathbb{R}^d \to \mathbb{R}^K$ | neural network classifier | $\beta_U$ | upper bound of minimum distortion |
| $\boldsymbol{x_0} \in \mathbb{R}^d$ | original input vector | $L_q^j$ | Lipschitz constant |
| $\boldsymbol{x_a} \in \mathbb{R}^d$ | adversarial example | $L_{q,x_0}^j$ | local Lipschitz constant |
| $\boldsymbol{\delta} \in \mathbb{R}^d$ | distortion $:= \boldsymbol{x_a} - \boldsymbol{x_0}$ | $B_p(\boldsymbol{x_0}, R)$ | hyper-ball with center $\boldsymbol{x_0}$ and radius $R$ |
| $\|\boldsymbol{\delta}\|_p$ | $\ell_p$ norm of distortion, $p \geq 1$ | CDF | cumulative distribution function |

intuitive manner[1]. Furthermore, our robustness analysis can be easily extended to non-differentiable classification functions (e.g. neural networks with ReLU) as in Lemma 3.3, whereas the analysis in Hein & Andriushchenko (2017) is restricted to differentiable functions. Specifically, Corollary 3.2.1 shows that the robustness analysis in (Hein & Andriushchenko, 2017) is in fact a special case of our analysis. We start our analysis by defining the notion of adversarial examples, minimum $\ell_p$ distortions, and lower/upper bounds. All the notations are summarized in Table 1.

**Definition 3.1** (perturbed example and adversarial example). Let $\boldsymbol{x_0} \in \mathbb{R}^d$ be an input vector of a $K$-class classification function $f : \mathbb{R}^d \to \mathbb{R}^K$ and the prediction is given as $c(\boldsymbol{x_0}) = \text{argmax}_{1 \leq i \leq K} f_i(\boldsymbol{x_0})$. Given $\boldsymbol{x_0}$, we say $\boldsymbol{x_a}$ is a *perturbed example* of $\boldsymbol{x_0}$ with noise $\boldsymbol{\delta} \in \mathbb{R}^d$ and $\ell_p$-distortion $\Delta_p$ if $\boldsymbol{x_a} = \boldsymbol{x_0} + \boldsymbol{\delta}$ and $\Delta_p = \|\boldsymbol{\delta}\|_p$. An *adversarial example* is a perturbed example $\boldsymbol{x_a}$ that changes $c(\boldsymbol{x_0})$. A successful *untargeted attack* is to find a $\boldsymbol{x_a}$ such that $c(\boldsymbol{x_a}) \neq c(\boldsymbol{x_0})$ while a successful *targeted attack* is to find a $\boldsymbol{x_a}$ such that $c(\boldsymbol{x_a}) = t$ given a target class $t \neq c(\boldsymbol{x_0})$.

**Definition 3.2** (minimum adversarial distortion $\Delta_{p,\min}$). Given an input vector $\boldsymbol{x_0}$ of a classifier $f$, the *minimum $\ell_p$ adversarial distortion* of $\boldsymbol{x_0}$, denoted as $\Delta_{p,\min}$, is defined as the smallest $\Delta_p$ over all adversarial examples of $\boldsymbol{x_0}$.

**Definition 3.3** (lower bound of $\Delta_{p,\min}$). Suppose $\Delta_{p,\min}$ is the minimum adversarial distortion of $\boldsymbol{x_0}$. A lower bound of $\Delta_{p,\min}$, denoted by $\beta_L$ where $\beta_L \leq \Delta_{p,\min}$, is defined such that any perturbed examples of $\boldsymbol{x_0}$ with $\|\boldsymbol{\delta}\|_p \leq \beta_L$ are not adversarial examples.

**Definition 3.4** (upper bound of $\Delta_{p,\min}$). Suppose $\Delta_{p,\min}$ is the minimum adversarial distortion of $\boldsymbol{x_0}$. An upper bound of $\Delta_{p,\min}$, denoted by $\beta_U$ where $\beta_U \geq \Delta_{p,\min}$, is defined such that there exists an adversarial example of $\boldsymbol{x_0}$ with $\|\boldsymbol{\delta}\|_p \geq \beta_U$.

The lower and upper bounds are instance-specific because they depend on the input $\boldsymbol{x_0}$. While $\beta_U$ can be easily given by finding an adversarial example of $\boldsymbol{x_0}$ using any attack method, $\beta_L$ is not easy to find. $\beta_L$ guarantees that the classifier is robust to any perturbations with $\|\delta\|_p \leq \beta_L$, certifying the robustness of the classifier. Below we show how to derive a formal robustness guarantee of a classifier with Lipschitz continuity assumption. Specifically, our analysis obtains a lower bound of $\ell_p$ minimum adversarial distortion $\beta_L = \min_{j \neq c} \frac{f_c(\boldsymbol{x_0}) - f_j(\boldsymbol{x_0})}{L_q^j}$.

**Lemma 3.1** (Lipschitz continuity and its relationship with gradient norm (Paulavičius & Žilinskas, 2006)). *Let $S \subset \mathbb{R}^d$ be a convex bounded closed set and let $h(\boldsymbol{x}) : S \to \mathbb{R}$ be a continuously differentiable function on an open set containing $S$. Then, $h(\boldsymbol{x})$ is a Lipschitz function with Lipschitz constant $L_q$ if the following inequality holds for any $\boldsymbol{x}, \boldsymbol{y} \in S$:*

$$|h(\boldsymbol{x}) - h(\boldsymbol{y})| \leq L_q \|\boldsymbol{x} - \boldsymbol{y}\|_p, \tag{1}$$

*where $L_q = \max\{\|\nabla h(\boldsymbol{x})\|_q : \boldsymbol{x} \in S\}, \nabla h(\boldsymbol{x}) = (\frac{\partial h(\boldsymbol{x})}{\partial x_1}, \cdots, \frac{\partial h(\boldsymbol{x})}{\partial x_d})^\top$ is the gradient of $h(\boldsymbol{x})$, and $\frac{1}{p} + \frac{1}{q} = 1, 1 \leq p, q \leq \infty$.*

Given Lemma 3.1, we then provide a formal guarantee to the lower bound $\beta_L$.

**Theorem 3.2** (Formal guarantee on lower bound $\beta_L$ for untargeted attack). *Let $\boldsymbol{x_0} \in \mathbb{R}^d$ and $f : \mathbb{R}^d \to \mathbb{R}^K$ be a multi-class classifier with continuously differentiable components $f_i$ and let $c = \text{argmax}_{1 \leq i \leq K} f_i(\boldsymbol{x_0})$ be the class which $f$ predicts for $\boldsymbol{x_0}$. For all $\boldsymbol{\delta} \in \mathbb{R}^d$ with*

$$\|\boldsymbol{\delta}\|_p \leq \min_{j \neq c} \frac{f_c(\boldsymbol{x_0}) - f_j(\boldsymbol{x_0})}{L_q^j}, \tag{2}$$

---

[1] The authors in Hein & Andriushchenko (2017) implicitly assume Lipschitz continuity and use Mean Value Theorem and Hölder's Inequality to prove their main theorem. Here we provide a simple and direct proof with Lipschitz continuity assumption and without involving Mean Value Theorem and Hölder's Inequality.

$\text{argmax}_{1 \le i \le K} f_i(\boldsymbol{x_0} + \boldsymbol{\delta}) = c$ holds with $\frac{1}{p} + \frac{1}{q} = 1$, $1 \le p, q \le \infty$ and $L_q^j$ is the Lipschitz constant for the function $f_c(\boldsymbol{x}) - f_j(\boldsymbol{x})$ in $\ell_p$ norm. In other words, $\beta_L = \min_{j \ne c} \frac{f_c(\boldsymbol{x_0}) - f_j(\boldsymbol{x_0})}{L_q^j}$ is a lower bound of minimum distortion.

The intuitions behind Theorem 3.2 is shown in Figure 1 with an one-dimensional example. The function value $g(x) = f_c(x) - f_j(x)$ near point $x_0$ is inside a double cone formed by two lines passing $(x_0, g(x_0))$ and with slopes equal to $\pm L_q$, where $L_q$ is the (local) Lipschitz constant of $g(x)$ near $x_0$. In other words, the function value of $g(x)$ around $x_0$, i.e. $g(x_0 + \delta)$ can be bounded by $g(x_0)$, $\delta$ and the Lipschitz constant $L_q$. When $g(x_0 + \delta)$ is decreased to 0, an adversarial example is found and the minimal change of $\delta$ is $\frac{g(x_0)}{L_q}$. The complete proof is deferred to Appendix A.

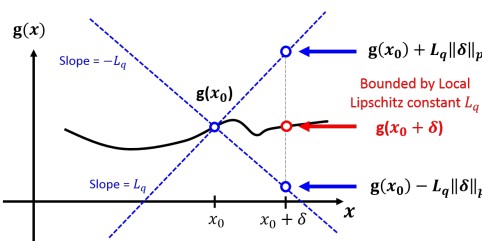

Figure 1: Intuitions behind Theorem 3.2.

**Remark 1.** *$L_q^j$ is the Lipschitz constant of the function involving cross terms: $f_c(\boldsymbol{x}) - f_j(\boldsymbol{x})$, hence we also call it cross Lipschitz constant following (Hein & Andriushchenko, 2017).*

To distinguish our analysis from (Hein & Andriushchenko, 2017), we show in Corollary 3.2.1 that we can obtain the same result in (Hein & Andriushchenko, 2017) by Theorem 3.2. In fact, the analysis in (Hein & Andriushchenko, 2017) is a special case of our analysis because the authors implicitly assume Lipschitz continuity on $f_i(\boldsymbol{x})$ when requiring $f_i(\boldsymbol{x})$ to be continuously differentiable. They use local Lipschitz constant ($L_{q,x_0}$) instead of global Lipschitz constant ($L_q$) to obtain a tighter bound in the adversarial perturbation $\boldsymbol{\delta}$.

**Corollary 3.2.1** (Formal guarantee on $\beta_L$ for untargeted attack). [2] *Let $L_{q,x_0}^j$ be local Lipschitz constant of function $f_c(\boldsymbol{x}) - f_j(\boldsymbol{x})$ at $\boldsymbol{x_0}$ over some fixed ball $B_p(\boldsymbol{x_0}, R) := \{\boldsymbol{x} \in \mathbb{R}^d \mid \|\boldsymbol{x} - \boldsymbol{x_0}\|_p \le R\}$ and let $\boldsymbol{\delta} \in B_p(\boldsymbol{0}, R)$. By Theorem 3.2, we obtain the bound in (Hein & Andriushchenko, 2017):*

$$\|\boldsymbol{\delta}\|_p \le \min \left\{ \min_{j \ne c} \frac{f_c(\boldsymbol{x_0}) - f_j(\boldsymbol{x_0})}{L_{q,x_0}^j}, R \right\}. \tag{3}$$

An important use case of Theorem 3.2 and Corollary 3.2.1 is the bound for targeted attack:

**Corollary 3.2.2** (Formal guarantee on $\beta_L$ for targeted attack). *Assume the same notation as in Theorem 3.2 and Corollary 3.2.1. For a specified target class $j$, we have $\|\boldsymbol{\delta}\|_p \le \min \left\{ \frac{f_c(\boldsymbol{x_0}) - f_j(\boldsymbol{x_0})}{L_{q,x_0}^j}, R \right\}$.*

In addition, we further extend Theorem 3.2 to a special case of *non-differentiable* functions – neural networks with ReLU activations. In this case the Lipchitz constant used in Lemma 3.1 can be replaced by the maximum norm of directional derivative, and our analysis above will go through.

**Lemma 3.3** (Formal guarantee on $\beta_L$ for ReLU networks). [3] *Let $h(\cdot)$ be a l-layer ReLU neural network with $W_i$ as the weights for layer $i$. We ignore bias terms as they don't contribute to gradient.*

$$h(\boldsymbol{x}) = \sigma(W_l \sigma(W_{l-1} \dots \sigma(W_1 \boldsymbol{x})))$$

*where $\sigma(u) = \max(0, u)$. Let $S \subset \mathbb{R}^d$ be a convex bounded closed set, then equation (1) holds with $L_q = \sup_{\boldsymbol{x} \in S} \{|\sup_{\|\boldsymbol{d}\|_p = 1} D^+ h(\boldsymbol{x}; \boldsymbol{d})|\}$ where $D^+ h(\boldsymbol{x}; \boldsymbol{d}) := \lim_{t \to 0^+} \frac{h(\boldsymbol{x} + t\boldsymbol{d}) - h(\boldsymbol{x})}{t}$ is the one-sided directional direvative, then Theorem 3.2, Corollary 3.2.1 and Corollary 3.2.2 still hold.*

## 4 THE CLEVER ROBUSTNESS METRIC VIA EXTREME VALUE THEORY

In this section, we provide an algorithm to compute the robustness metric CLEVER with the aid of extreme value theory, where CLEVER can be viewed as an efficient estimator of the lower bound $\beta_L$ and is the first attack-agnostic score that applies to any neural network classifiers. Recall in Section 3

---

[2] proof deferred to Appendix B    [3] proof deferred to Appendix C

we show that the lower bound of network robustness is associated with $g(\boldsymbol{x_0})$ and its cross Lipschitz constant $L_{q,x_0}^j$, where $g(\boldsymbol{x_0}) = f_c(\boldsymbol{x_0}) - f_j(\boldsymbol{x_0})$ is readily available at the output of a classifier and $L_{q,x_0}^j$ is defined as $\max_{\boldsymbol{x} \in B_p(\boldsymbol{x_0}, R)} \|\nabla g(\boldsymbol{x})\|_q$. Although $\nabla g(\boldsymbol{x})$ can be calculated easily via back propagation, computing $L_{q,x_0}^j$ is more involved because it requires to obtain the maximum value of $\|\nabla g(\boldsymbol{x})\|_q$ in a ball. Exhaustive search on low dimensional $\boldsymbol{x}$ in $B_p(\boldsymbol{x}_0, R)$ seems already infeasible, not to mention the image classifiers with large feature dimensions of our interest. For instance, the feature dimension $d = 784, 3072, 150528$ for MNIST, CIFAR and ImageNet respectively.

One approach to compute $L_{q,x_0}^j$ is through sampling a set of points $\boldsymbol{x}^{(i)}$ in a ball $B_p(\boldsymbol{x}_0, R)$ around $\boldsymbol{x}_0$ and taking the maximum value of $\|\nabla g(\boldsymbol{x}^{(i)})\|_q$. However, a significant amount of samples might be needed to obtain a good estimate of $\max \|\nabla g(\boldsymbol{x})\|_q$ and it is unknown how good the estimate is compared to the true maximum. Fortunately, Extreme Value Theory ensures that the maximum value of random variables can only follow one of the three extreme value distributions, which is useful to estimate $\max \|\nabla g(\boldsymbol{x})\|_q$ with only a tractable number of samples.

It is worth noting that although Wood & Zhang (1996) also applied extreme value theory to estimate the Lipschitz constant. However, there are two main differences between their work and this paper. First of all, the sampling methodology is entirely different. Wood & Zhang (1996) calculates the slopes between pairs of sample points whereas we directly take samples on the norm of gradient as in Lemma 3.1. Secondly, the functions considered in Wood & Zhang (1996) are only one-dimensional as opposed to the high-dimensional classification functions considered in this paper. For comparison, we show in our experiment that the approach in Wood & Zhang (1996), denoted as SLOPE in Table 3 and Figure 4, perform poorly for high-dimensional classifiers such as deep neural networks.

## 4.1 ESTIMATE $L_{q,x_0}^j$ VIA EXTREME VALUE THEORY

When sampling a point $\boldsymbol{x}$ uniformly in $B_p(\boldsymbol{x}_0, R)$, $\|\nabla g(\boldsymbol{x})\|_q$ can be viewed as a random variable characterized by a cumulative distribution function (CDF). For the purpose of illustration, we derived the CDF for a 2-layer neural network in Theorem D.1.[4] For any neural networks, suppose we have $n$ samples $\{\|\nabla g(\boldsymbol{x}^{(i)})\|_q\}$, and denote them as a sequence of independent and identically distributed (iid) random variables $Y_1, Y_2, \cdots, Y_n$, each with CDF $F_Y(y)$. The CDF of $\max\{Y_1, \cdots, Y_n\}$, denoted as $F_Y^n(y)$, is called the limit distribution of $F_Y(y)$. Fisher-Tippett-Gnedenko theorem says that $F_Y^n(y)$, if exists, can only be one of the three family of extreme value distributions – the Gumbel class, the Fréchet class and the reverse Weibull class.

**Theorem 4.1** (Fisher-Tippett-Gnedenko Theorem). *If there exists a sequence of pairs of real numbers $(a_n, b_n)$ such that $a_n > 0$ and $\lim_{n \to \infty} F_Y^n(a_n y + b_n) = G(y)$, where $G$ is a non-degenerate distribution function, then $G$ belongs to either the Gumbel class (Type I), the Fréchet class (Type II) or the Reverse Weibull class (Type III) with their CDFs as follows:*

$$\text{Gumbel class (Type I):} \quad G(y) = \exp\left\{-\exp\left[-\frac{y - a_W}{b_W}\right]\right\}, \quad y \in \mathbb{R},$$

$$\text{Fréchet class (Type II):} \quad G(y) = \begin{cases} 0, & \text{if } y < a_W, \\ \exp\{-\left(\frac{y - a_W}{b_W}\right)^{-c_W}\}, & \text{if } y \geq a_W, \end{cases}$$

$$\text{Reverse Weibull class (Type III):} \quad G(y) = \begin{cases} \exp\{-\left(\frac{a_W - y}{b_W}\right)^{c_W}\}, & \text{if } y < a_W, \\ 1, & \text{if } y \geq a_W, \end{cases}$$

*where $a_W \in \mathbb{R}$, $b_W > 0$ and $c_W > 0$ are the location, scale and shape parameters, respectively.*
Theorem 4.1 implies that the maximum values of the samples follow one of the three families of distributions. If $g(\boldsymbol{x})$ has a bounded Lipschitz constant, $\|\nabla g(\boldsymbol{x}^{(i)})\|_q$ is also bounded, thus its limit distribution must have a finite right end-point. We are particularly interested in the reverse Weibull class, as its CDF has a finite right end-point (denoted as $a_W$). The right end-point reveals the upper limit of the distribution, known as the *extreme value*. The extreme value is exactly the unknown local cross Lipschitz constant $L_{q,\boldsymbol{x}_0}^j$ we would like to estimate in this paper. To estimate $L_{q,\boldsymbol{x}_0}^j$, we first generate $N_s$ samples of $\boldsymbol{x}^{(i)}$ over a fixed ball $B_p(\boldsymbol{x_0}, R)$ uniformly and independently in each batch with a total of $N_b$ batches. We then compute $\|\nabla g(\boldsymbol{x}^{(i)})\|_q$ and store the maximum values of each batch in set $S$. Next, with samples in $S$, we perform a maximum likelihood estimation of reverse Weibull distribution parameters, and the location estimate $\hat{a}_W$ is used as an estimate of $L_{q,\boldsymbol{x}_0}^j$.

---

[4] The theorem and proof are deferred to Appendix D.

## 4.2 COMPUTE CLEVER: A ROBUSTNESS SCORE OF NEURAL NETWORK CLASSIFIERS

Given an instance $x_0$, its classifier $f(x_0)$ and a target class $j$, a targeted CLEVER score of the classifier's robustness can be computed via $g(x_0)$ and $L_{q,x_0}^j$. Similarly, untargeted CLEVER scores can be computed. With the proposed procedure of estimating $L_{q,x_0}^j$ described in Section 4.1, we summarize the flow of computing CLEVER score for both targeted attacks and un-targeted attacks in Algorithm 1 and 2, respectively.

---

**Algorithm 1:** `CLEVER-t`, compute **CLEVER** score for **t**argeted attack

**Input:** a $K$-class classifier $f(x)$, data example $x_0$ with predicted class $c$, target class $j$, batch size $N_b$, number of samples per batch $N_s$, perturbation norm $p$, maximum perturbation $R$
**Result:** CLEVER Score $\mu \in \mathbb{R}_+$ for target class $j$
1  $S \leftarrow \{\emptyset\}$, $g(x) \leftarrow f_c(x) - f_j(x)$, $q \leftarrow \frac{p}{p-1}$.
2  **for** $i \leftarrow 1$ **to** $N_b$ **do**
3  $\quad$ **for** $k \leftarrow 1$ **to** $N_s$ **do**
4  $\quad\quad$ randomly select a point $x^{(i,k)} \in B_p(x_0, R)$
5  $\quad\quad$ compute $b_{ik} \leftarrow \|\nabla g(x^{(i,k)})\|_q$ via back propagation
6  $\quad$ **end**
7  $\quad$ $S \leftarrow S \cup \{\max_k\{b_{ik}\}\}$
8  **end**
9  $\hat{a}_W \leftarrow$ MLE of location parameter of reverse Weibull distribution on $S$
10 $\mu \leftarrow \min(\frac{g(x_0)}{\hat{a}}, R)$

---

**Algorithm 2:** `CLEVER-u`, compute **CLEVER** score for **u**n-targeted attack

**Input:** Same as Algorithm 1, but without a target class $j$
**Result:** CLEVER score $\nu \in \mathbb{R}_+$ for un-targeted attack
1  **for** $j \leftarrow 1$ **to** $K$, $j \neq c$ **do**
2  $\quad$ $\mu_j \leftarrow$ `CLEVER-t`$(f, x_0, c, j, N_b, N_s, p, R)$
3  **end**
4  $\nu \leftarrow \min_j\{\mu_j\}$

---

## 5 EXPERIMENTAL RESULTS

### 5.1 NETWORKS AND PARAMETER SETUP

We conduct experiments on CIFAR-10 (CIFAR for short), MNIST, and ImageNet data sets. For the former two smaller datasets CIFAR and MNIST, we evaluate CLEVER scores on four relatively small networks: a single hidden layer MLP with softplus activation (with the same number of hidden units as in (Hein & Andriushchenko, 2017)), a 7-layer AlexNet-like CNN (with the same structure as in (Carlini & Wagner, 2017b)), and the 7-layer CNN with defensive distillation (Papernot et al., 2016) (DD) and bounded ReLU (Zantedeschi et al., 2017) (BReLU) defense techniques employed.

For ImageNet data set, we use three popular deep network architectures: a 50-layer Residual Network (He et al., 2016) (ResNet-50), Inception-v3 (Szegedy et al., 2016) and MobileNet (Howard et al., 2017). They were chosen for the following reasons: (i) they all yield (close to) state-of-the-art performance among equal-sized networks; and (ii) their architectures are significantly different with unique building blocks, i.e., residual block in ResNet, inception module in Inception net, and depthwise separable convolution in MobileNet. Therefore, their diversity in network architectures is appropriate to test our robustness metric. For MobileNet, we set the width multiplier to 1.0, achieving a 70.6% accuracy on ImageNet. We used public pretrained weights for all ImageNet models[5].

In all our experiments, we set the sampling parameters $N_b = 500$, $N_s = 1024$ and $R = 5$. For targeted attacks, we use 500 test-set images for CIFAR and MNIST and use 100 test-set images for ImageNet; for each image, we evaluate its targeted CLEVER score for three targets: a random target class, a least likely class (the class with lowest probability when predicting the original example),

---

[5] Pretrained models can be downloaded at https://github.com/tensorflow/models/tree/master/research/slim

and the top-2 class (the class with largest probability except for the true class, which is usually the easiest target to attack). We also conduct untargeted attacks on MNIST and CIFAR for 100 test-set images, and evaluate their untargeted CLEVER scores. Our experiment code is publicly available[6].

## 5.2 FITTING GRADIENT NORM SAMPLES WITH REVERSE WEIBULL DISTRIBUTIONS

We fit the cross Lipschitz constant samples in $S$ (see Algorithm 1) with reverse Weibull class distribution to obtain the maximum likelihood estimate of the location parameter $\hat{a}_W$, scale parameter $\hat{b}_W$ and shape parameter $\hat{c}_W$, as introduced in Theorem 4.1. To validate that reverse Weibull distribution is a good fit to the empirical distribution of the cross Lipschitz constant samples, we conduct Kolmogorov-Smirnov goodness-of-fit test (a.k.a. K-S test) to calculate the K-S test statistics $D$ and corresponding $p$-values. The null hypothesis is that samples $S$ follow a reverse Weibull distribution.

Figure 2 plots the probability distribution function of the cross Lipschitz constant samples and the fitted Reverse Weibull distribution for images from various data sets and network architectures. The estimated MLE parameters, $p$-values, and the K-S test statistics $D$ are also shown. We also calculate the percentage of examples whose estimation have $p$-values greater than 0.05, as illustrated in Figure 3. If the $p$-value is greater than 0.05, the null hypothesis cannot be rejected, meaning that the underlying data samples fit a reverse Weibull distribution well. Figure 3 shows that all numbers are close to 100%, validating the use of reverse Weibull distribution as an underlying distribution of gradient norm samples empirically. Therefore, the fitted location parameter of reverse Weibull distribution (i.e., the extreme value), $\hat{a}_W$, can be used as a good estimation of local cross Lipschitz constant to calculate the CLEVER score. The exact numbers are shown in Table 5 in Appendix E.

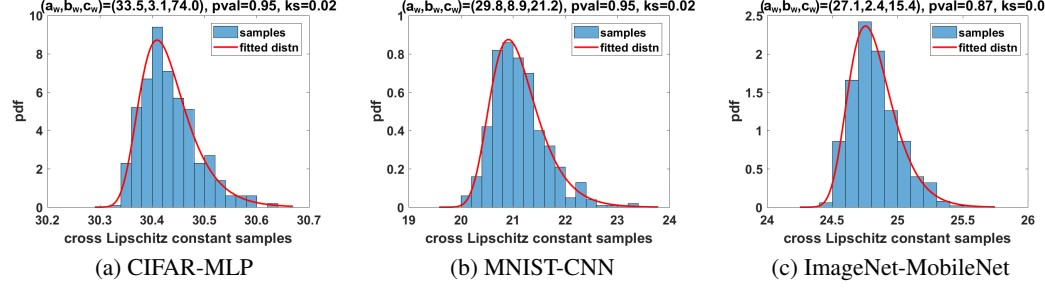

(a) CIFAR-MLP     (b) MNIST-CNN     (c) ImageNet-MobileNet

Figure 2: The cross Lipschitz constant samples for three images from CIFAR, MNIST and ImageNet datasets, and their fitted Reverse Weibull distributions with the corresponding MLE estimates of location, scale and shape parameters $(a_W, b_W, c_W)$ shown on the top of each plot. The $D$-statistics of K-S test and p-values are denoted as *ks* and *pval*. With small *ks* and high p-value, the hypothesized reverse Weibull distribution fits the empirical distribution of cross Lipschitz constant samples well.

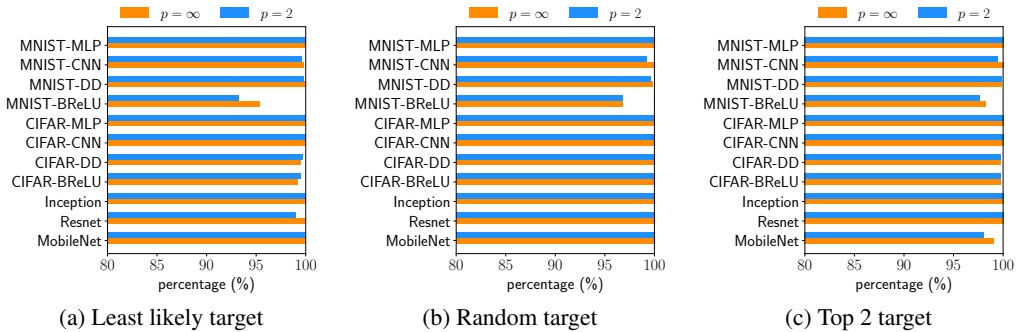

(a) Least likely target     (b) Random target     (c) Top 2 target

Figure 3: The percentage of examples whose null hypothesis (the samples $S$ follow a reverse Weibull distribution) cannot be rejected by K-S test with a significance level of 0.05 for $p = 2$ and $p = \infty$. All numbers for each model are close to 100%, indicating $S$ fits reverse Weibull distributions well.

---

[6] Source code is available at https://github.com/huanzhang12/CLEVER

## 5.3 COMPARING CLEVER SCORE WITH ATTACK-SPECIFIC NETWORK ROBUSTNESS

We apply the state-of-the-art white-box attack methods, iterative fast gradient sign method (I-FGSM) (Goodfellow et al., 2015; Kurakin et al., 2016b) and Carlini and Wagner's attack (CW) (Carlini & Wagner, 2017b), to find adversarial examples for 11 networks, including 4 networks trained on CIFAR, 4 networks trained on MNIST, and 3 networks trained on ImageNet. For CW attack, we run 1000 iterations for ImageNet and CIFAR, and 2000 iterations for MNIST, as MNIST has shown to be more difficult to attack (Chen et al., 2017b). Attack learning rate is individually tuned for each model: 0.001 for Inception-v3 and ResNet-50, 0.0005 for MobileNet and 0.01 for all other networks. For I-FGSM, we run 50 iterations and choose the optimal $\epsilon \in \{0.01, 0.025, 0.05, 0.1, 0.3, 0.5, 0.8, 1.0\}$ to achieve the smallest $\ell_\infty$ distortion for each individual image. For defensively distilled (DD) networks, 50 iterations of I-FGSM are not sufficient; we use 250 iterations for CIFAR-DD and 500 iterations for MNIST-DD to achieve a 100% success rate. For the problem to be non-trivial, images that are classified incorrectly are skipped. We report 100% attack success rates for all the networks, and thus the average distortion of adversarial examples can indicate the attack-specific robustness of each network. For comparison, we compute the CLEVER scores for the same set of images and attack targets. To the best of our knowledge, CLEVER is the first attack-independent robustness score that is capable of handling the large networks studied in this paper, so we directly compare it with the attack-induced distortion metrics in our study.

We evaluate the effectiveness of our CLEVER score by comparing the upper bound $\beta_U$ (found by attacks) and CLEVER score, where CLEVER serves as an estimated lower bound, $\beta_L$. Table 3 compares the average $\ell_2$ and $\ell_\infty$ distortions of adversarial examples found by targeted CW and I-FGSM attacks and the corresponding average *targeted* CLEVER scores for $\ell_2$ and $\ell_\infty$ norms, and Figure 4 visualizes the results for $\ell_\infty$ norm. Similarly, Table 2 compares untargeted CW and I-FGSM attacks with *untargeted* CLEVER scores. As expected, CLEVER is smaller than the distortions of adversarial images in most cases. More importantly, since CLEVER is independent of attack algorithms, the reported CLEVER scores can roughly indicate the distortion of the *best possible* attack in terms of a specific $\ell_p$ distortion. The average $\ell_2$ distortion found by CW attack is close to the $\ell_2$ CLEVER score, indicating CW is a strong $\ell_2$ attack. In addition, when a defense mechanism (Defensive Distillation or Bounded ReLU) is used, the corresponding CLEVER scores are consistently increased (except for CIFAR-BReLU), indicating that the network is indeed made more resilient to adversarial perturbations. For CIFAR-BReLU, both CLEVER scores and $\ell_p$ norm of adversarial examples found by CW attack decrease, implying that bound ReLU is an ineffective defense for CIFAR. CLEVER scores can be seen as a security checkpoint for unseen attacks. For example, if there is a substantial gap in distortion between the CLEVER score and the considered attack algorithms, it may suggest the existence of a more effective attack that can close the gap.

Since CLEVER score is derived from an estimation of the robustness lower bound, we further verify the viability of CLEVER per each example, i.e., whether it is usually smaller than the upper bound found by attacks. Table 4 shows the percentage of inaccurate estimations where the CLEVER score is larger than the distortion of adversarial examples found by CW and I-FGSM attacks in three ImageNet networks. We found that CLEVER score provides an accurate estimation for most of the examples. For MobileNet and Resnet-50, our CLEVER score is a strict lower bound of these two attacks for more than 96% of tested examples. For Inception-v3, the condition of strict lower bound

Table 2: Comparison between the average untargeted CLEVER score and distortion found by CW and I-FGSM untargeted attacks. DD and BReLU represent Defensive Distillation and Bounded ReLU defending methods applied to the baseline CNN network.

|  | CW | | I-FGSM | | CLEVER | |
|---|---|---|---|---|---|---|
|  | $\ell_2$ | $\ell_\infty$ | $\ell_2$ | $\ell_\infty$ | $\ell_2$ | $\ell_\infty$ |
| MNIST-MLP | 1.113 | 0.215 | 3.564 | 0.178 | 0.819 | 0.041 |
| MNIST-CNN | 1.500 | 0.455 | 4.439 | 0.288 | 0.721 | 0.057 |
| MNIST-DD | 1.548 | 0.409 | 5.617 | 0.283 | 0.865 | 0.063 |
| MNIST-BReLU | 1.337 | 0.433 | 3.851 | 0.285 | 0.833 | 0.065 |
| CIFAR-MLP | 0.253 | 0.018 | 0.885 | 0.016 | 0.219 | 0.005 |
| CIFAR-CNN | 0.195 | 0.023 | 0.721 | 0.018 | 0.072 | 0.002 |
| CIFAR-DD | 0.285 | 0.032 | 1.136 | 0.024 | 0.130 | 0.004 |
| CIFAR-BReLU | 0.159 | 0.019 | 0.519 | 0.013 | 0.045 | 0.001 |

Table 3: Comparison of the average targeted CLEVER scores with average $\ell_\infty$ and $\ell_2$ distortions found by CW, I-FSGM attacks, and the average scores calculated by using the algorithm in Wood & Zhang (1996) (denoted as SLOPE) to estimate Lipschitz constant. DD and BReLU denote Defensive Distillation and Bounded ReLU defending methods applied to the CNN network. We did not include SLOPE in ImageNet networks because it has been shown to be ineffective even for smaller networks.

(a) avergage $\ell_\infty$ distortion of CW and I-FGSM targeted attacks, and CLEVER and SLOPE estimation. Some very large SLOPE estimates (in parentheses) exceeding the maximum possible $\ell_\infty$ distortion are reported as 1.

| | Least Likely Target | | | | Random Target | | | | Top-2 Target | | | |
|---|---|---|---|---|---|---|---|---|---|---|---|---|
| | CW | I-FGSM | CLEVER | SLOPE | CW | I-FGSM | CLEVER | SLOPE | CW | I-FGSM | CLEVER | SLOPE |
| MNIST-MLP | 0.475 | 0.223 | 0.071 | 0.808 | 0.337 | 0.173 | 0.072 | 0.813 | 0.218 | 0.119 | 0.069 | 0.786 |
| MNIST-CNN | 0.601 | 0.313 | 0.090 | 0.996 | 0.550 | 0.264 | 0.088 | 0.982 | 0.451 | 0.211 | 0.070 | 0.826 |
| MNIST-DD | 0.578 | 0.283 | 0.103 | 1 (1.090) | 0.531 | 0.238 | 0.091 | 0.953 | 0.412 | 0.165 | 0.091 | 0.984 |
| MNIST-BReLU | 0.601 | 0.276 | 0.257 | 1 (5.327) | 0.544 | 0.238 | 0.187 | 3.907 | 0.442 | 0.196 | 0.117 | 1 (2.470) |
| CIFAR-MLP | 0.086 | 0.039 | 0.014 | 0.294 | 0.051 | 0.024 | 0.014 | 0.284 | 0.019 | 0.013 | 0.014 | 0.286 |
| CIFAR-CNN | 0.053 | 0.033 | 0.005 | 0.153 | 0.042 | 0.023 | 0.005 | 0.148 | 0.022 | 0.013 | 0.004 | 0.129 |
| CIFAR-DD | 0.091 | 0.053 | 0.011 | 0.278 | 0.066 | 0.032 | 0.010 | 0.255 | 0.033 | 0.014 | 0.007 | 0.184 |
| CIFAR-BReLU | 0.045 | 0.030 | 0.004 | 0.250 | 0.034 | 0.022 | 0.003 | 0.173 | 0.018 | 0.012 | 0.002 | 0.095 |
| Inception-v3 | 0.023 | 0.011 | 0.002 | - | 0.021 | 0.012 | 0.002 | - | 0.010 | 0.011 | 0.001 | - |
| Resnet-50 | 0.031 | 0.015 | 0.002 | - | 0.025 | 0.012 | 0.002 | - | 0.010 | 0.010 | 0.001 | - |
| MobileNet | 0.025 | 0.010 | 0.003 | - | 0.018 | 0.010 | 0.002 | - | 0.006 | 0.010 | 0.001 | - |

(b) average $\ell_2$ distortion of CW and I-FGSM targeted attacks, and CLEVER and SLOPE estimation. Some very large SLOPE estimates (in parentheses) exceeding the sampling radius $R = 5$ are reported as 5.

| | Least Likely Target | | | | Random Target | | | | Top-2 Target | | | |
|---|---|---|---|---|---|---|---|---|---|---|---|---|
| | CW | I-FGSM | CLEVER | SLOPE | CW | I-FGSM | CLEVER | SLOPE | CW | I-FGSM | CLEVER | SLOPE |
| MNIST-MLP | 2.575 | 4.273 | 1.409 | 5 (8.028) | 1.833 | 3.369 | 1.432 | 5 (8.102) | 1.128 | 2.374 | 1.383 | 5 (7.853) |
| MNIST-CNN | 2.377 | 4.417 | 1.257 | 5 (9.947) | 2.005 | 3.902 | 1.227 | 5 (9.619) | 1.504 | 3.242 | 0.987 | 5 (7.921) |
| MNIST-DD | 2.644 | 4.957 | 1.532 | 5 (10.628) | 2.240 | 4.253 | 1.340 | 5 (9.493) | 1.542 | 3.010 | 1.330 | 5 (9.646) |
| MNIST-BReLU | 2.349 | 5.170 | 3.312 | 5 (52.058) | 1.923 | 4.544 | 2.565 | 5 (37.531) | 1.404 | 3.778 | 1.583 | 5 (23.548) |
| CIFAR-MLP | 1.123 | 1.896 | 0.620 | 5 (5.013) | 0.673 | 1.214 | 0.597 | 4.806 | 0.262 | 0.689 | 0.599 | 4.949 |
| CIFAR-CNN | 0.836 | 1.067 | 0.156 | 2.630 | 0.372 | 0.837 | 0.146 | 2.497 | 0.188 | 0.552 | 0.123 | 2.195 |
| CIFAR-DD | 2.065 | 1.540 | 0.347 | 4.735 | 0.624 | 1.097 | 0.307 | 4.279 | 0.296 | 0.582 | 0.220 | 3.083 |
| CIFAR-BReLU | 0.407 | 0.928 | 0.140 | 4.125 | 0.303 | 0.732 | 0.103 | 2.944 | 0.152 | 0.494 | 0.052 | 1.564 |
| Inception-v3 | 0.628 | 2.244 | 0.524 | - | 0.595 | 2.261 | 0.466 | - | 0.287 | 2.073 | 0.234 | - |
| Resnet-50 | 0.767 | 2.410 | 0.357 | - | 0.647 | 2.098 | 0.299 | - | 0.212 | 1.682 | 0.134 | - |
| MobileNet | 0.837 | 2.195 | 0.617 | - | 0.603 | 2.066 | 0.439 | - | 0.190 | 1.771 | 0.144 | - |

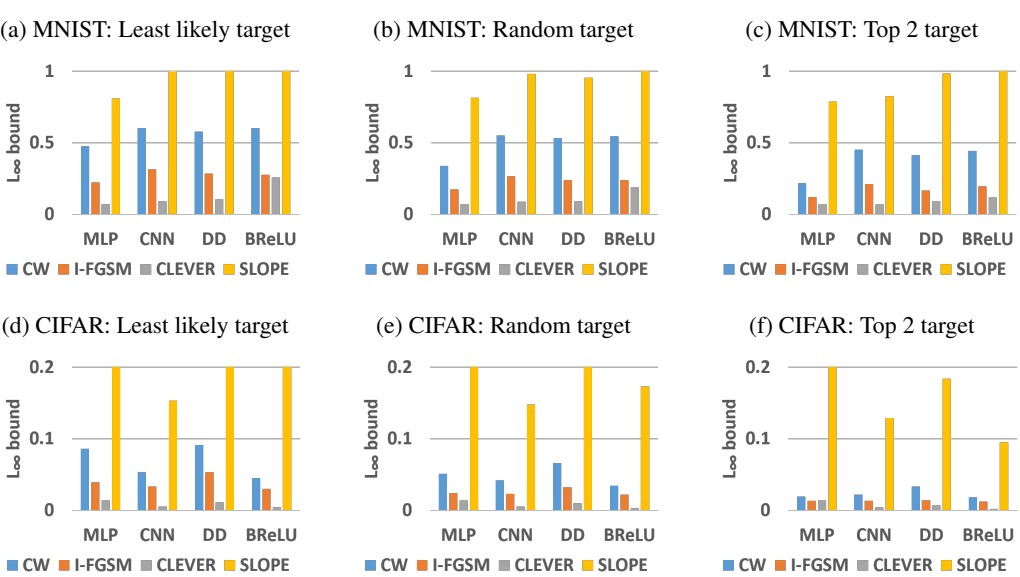

Figure 4: Comparison of $\ell_\infty$ distortion obtained by CW and I-FGSM attacks, CLEVER score and the slope based Lipschitz constant estimation (SLOPE) by Wood & Zhang (1996). SLOPE significantly exceeds the distortions found by attacks, thus it is an inappropriate estimation of lower bound $\beta_L$.

is worse (still more than 75%), but we found that in these cases the attack distortion only differs from our CLEVER score by a fairly small amount. In Figure 5 we show the empirical CDF of the gap between CLEVER score and the $\ell_2$ norm of adversarial distortion generated by CW attack for the same set of images in Table 4. In Figure 6, we plot the $\ell_2$ distortion and CLEVER scores for each

Table 4: Percentage of images in ImageNet where the CLEVER score for that image is greater than the adversarial distortion found by different attacks.

| | Least Likely Target | | | | Random Target | | | | Top-2 Target | | | |
| | CW | | I-FGSM | | CW | | I-FGSM | | CW | | I-FGSM | |
| | $L_2$ | $L_\infty$ | $L_2$ | $L_\infty$ | $L_2$ | $L_\infty$ | $L_2$ | $L_\infty$ | $L_2$ | $L_\infty$ | $L_2$ | $L_\infty$ |
|---|---|---|---|---|---|---|---|---|---|---|---|---|
| MobileNet | 4% | 0% | 0% | 0% | 2% | 0% | 0% | 0% | 0% | 0% | 0% | 0% |
| Resnet-50 | 4% | 0% | 0% | 0% | 2% | 0% | 0% | 0% | 1% | 0% | 0% | 0% |
| Inception-v3 | 25% | 0% | 0% | 0% | 23% | 0% | 0% | 0% | 15% | 0% | 0% | 0% |

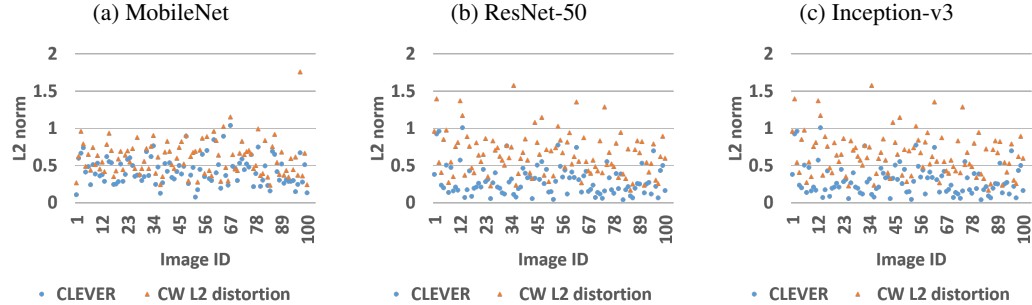

Figure 5: The empirical CDF of the gap between CLEVER score and the $\ell_2$ norm of adversarial distortion generated by CW attack with random targets for 100 images on 3 ImageNet networks.

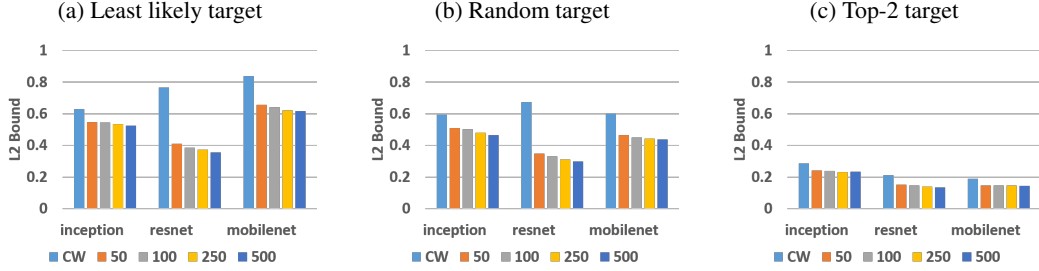

Figure 6: Comparison of the CLEVER scores (circle) and the $\ell_2$ norm of adversarial distortion generated by CW attack (triangle) with random targets for 100 images. The x-axis is image ID and the y-axis is the $\ell_2$ distortion metric.

Figure 7: Comparison of the CLEVER score calculated by $N_b = \{50, 100, 250, 500\}$ and the $\ell_2$ norm of adversarial distortion found by CW attack (CW) on 3 ImageNet models and 3 target types.

individual image. A positive gap indicates that CLEVER (estimated lower bound) is indeed less than the upper bound found by CW attack. Most images have a small positive gap, which signifies the near-optimality of CW attack in terms of $\ell_2$ distortion, as CLEVER suffices for an estimated capacity of the best possible attack.

## 5.4 TIME V.S. ESTIMATION ACCURACY

In Figure 7, we vary the number of samples ($N_b = 50, 100, 250, 500$) and compute the $\ell_2$ CLEVER scores for three large ImageNet models, Inception-v3, ResNet-50 and MobileNet. We observe that

50 or 100 samples are usually sufficient to obtain a reasonably accurate robustness estimation despite using a smaller number of samples. On a single GTX 1080 Ti GPU, the cost of 1 sample (with $N_s = 1024$) is measured as 2.9 s for MobileNet, 5.0 s for ResNet-50 and 8.9 s for Inception-v3, thus the computational cost of CLEVER is feasible for state-of-the-art large-scale deep neural networks. Additional figures for MNIST and CIFAR datasets are given in Appendix E.

## 6 CONCLUSION

In this paper, we propose the CLEVER score, a novel and generic metric to evaluate the robustness of a target neural network classifier to adversarial examples. Compared to the existing robustness evaluation approaches, our metric has the following advantages: (i) attack-agnostic; (ii) applicable to any neural network classifier; (iii) comes with strong theoretical guarantees; and (iv) is computationally feasible for large neural networks. Our extensive experiments show that the CLEVER score well matches the practical robustness indication of a wide range of natural and defended networks.

**Acknowledgment.** Luca Daniel and Tsui-Wei Weng are partially supported by MIT-Skoltech program and MIT-IBM Watson AI Lab. Cho-Jui Hsieh and Huan Zhang acknowledge the support of NSF via IIS-1719097.

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

APPENDIX

A    PROOF OF THEOREM 3.2

*Proof.* According to Lemma 3.1, the assumption that $g(\boldsymbol{x}) := f_c(\boldsymbol{x}) - f_j(\boldsymbol{x})$ is Lipschitz continuous with Lipschitz constant $L_q^j$ gives

$$|g(\boldsymbol{x}) - g(\boldsymbol{y})| \leq L_q^j \|\boldsymbol{x} - \boldsymbol{y}\|_p. \tag{4}$$

Let $\boldsymbol{x} = \boldsymbol{x_0} + \boldsymbol{\delta}$ and $\boldsymbol{y} = \boldsymbol{x_0}$ in (4), we get

$$|g(\boldsymbol{x_0} + \boldsymbol{\delta}) - g(\boldsymbol{x_0})| \leq L_q^j \|\boldsymbol{\delta}\|_p,$$

which can be rearranged into the following form

$$g(\boldsymbol{x_0}) - L_q^j \|\boldsymbol{\delta}\|_p \leq g(\boldsymbol{x_0} + \boldsymbol{\delta}) \leq g(\boldsymbol{x_0}) + L_q^j \|\boldsymbol{\delta}\|_p. \tag{5}$$

When $g(\boldsymbol{x_0} + \boldsymbol{\delta}) = 0$, an adversarial example is found. As indicated by (5), $g(\boldsymbol{x_0} + \boldsymbol{\delta})$ is lower bounded by $g(\boldsymbol{x_0}) - L_q^j \|\boldsymbol{\delta}\|_p$. If $\|\boldsymbol{\delta}\|_p$ is small enough such that $g(\boldsymbol{x_0}) - L_q^j \|\boldsymbol{\delta}\|_p \geq 0$, no adversarial examples can be found:

$$g(\boldsymbol{x_0}) - L_q^j \|\boldsymbol{\delta}\|_p \geq 0 \Rightarrow \|\boldsymbol{\delta}\|_p \leq \frac{g(\boldsymbol{x_0})}{L_q^j} \Rightarrow \|\boldsymbol{\delta}\|_p \leq \frac{f_c(\boldsymbol{x_0}) - f_j(\boldsymbol{x_0})}{L_q^j},$$

Finally, to achieve $\operatorname{argmax}_{1 \leq i \leq K} f_i(\boldsymbol{x_0} + \boldsymbol{\delta}) = c$, we take the minimum of the bound on $\|\boldsymbol{\delta}\|_p$ in (A) over $j \neq c$. I.e. if

$$\|\boldsymbol{\delta}\|_p \leq \min_{j \neq c} \frac{f_c(\boldsymbol{x_0}) - f_j(\boldsymbol{x_0})}{L_q^j},$$

the classifier decision can *never* be changed and the attack will *never* succeed. ☐

B    PROOF OF COROLLARY 3.2.1

*Proof.* By Lemma 3.1 and let $g = f_c - f_j$, we get $L_{q,x_0}^j = \max_{y \in B_p(x_0, R)} \|\nabla g(y)\|_q = \max_{y \in B_p(x_0, R)} \|\nabla f_j(y) - \nabla f_c(y)\|_q$, which then gives the bound in Theorem 2.1 of (Hein & Andriushchenko, 2017). ☐

C    PROOF OF LEMMA 3.3

*Proof.* For any $\boldsymbol{x}, \boldsymbol{y}$, let $\boldsymbol{d} = \frac{\boldsymbol{y} - \boldsymbol{x}}{\|\boldsymbol{y} - \boldsymbol{x}\|_p}$ be the unit vector pointing from $\boldsymbol{x}$ to $\boldsymbol{y}$ and $r = \|\boldsymbol{y} - \boldsymbol{x}\|_p$. Define uni-variate function $u(z) = h(\boldsymbol{x} + z\boldsymbol{d})$, then $u(0) = h(\boldsymbol{x})$ and $u(r) = h(\boldsymbol{y})$ and observe that $D^+ h(\boldsymbol{x} + z\boldsymbol{d}; \boldsymbol{d})$ and $D^+ h(\boldsymbol{x} + z\boldsymbol{d}; -\boldsymbol{d})$ are the right-hand and left-hand derivatives of $u(z)$, we have

$$u'(z) = \begin{cases} D^+ h(\boldsymbol{x} + z\boldsymbol{d}; \boldsymbol{d}) \leq L_q & \text{if } D^+ h(\boldsymbol{x} + z\boldsymbol{d}; \boldsymbol{d}) = D^+ h(\boldsymbol{x} + z\boldsymbol{d}; -\boldsymbol{d}) \\ \text{undefined} & \text{if } D^+ h(\boldsymbol{x} + z\boldsymbol{d}; \boldsymbol{d}) \neq D^+ h(\boldsymbol{x} + z\boldsymbol{d}; -\boldsymbol{d}) \end{cases}$$

For ReLU network, there can be at most finite number of points in $z \in (0, r)$ such that $g'(z)$ does not exist. This can be shown because each discontinuous $z$ is caused by some ReLU activation, and there are only finite combinations. Let $0 = z_0 < z_1 < \cdots < z_{k-1} < z_k = 1$ be those points. Then, using the fundamental theorem of calculus on each interval separately, there exists $\bar{z}_i \in (z_i, z_{i-1})$ for each $i$ such that

$$
\begin{aligned}
u(r) - u(0) &\leq \sum_{i=1}^k |u(z_i) - u(z_{i-1})| \\
&\leq \sum_{i=1}^k |u'(\bar{z}_i)(z_i - z_{i-1})| && \text{(Mean value theorem)} \\
&\leq \sum_{i=1}^k L_q |z_i - z_{i-1}|_p \\
&= L_q \|x - y\|_p. && (z_i \text{ are in line } (x, y))
\end{aligned}
$$

Theorem 3.2 and its corollaries remain valid after replacing Lemma 3.1 with Lemma 3.3. ☐

## D   THEOREM D.1 AND ITS PROOF

**Theorem D.1** ($F_Y(y)$ of one-hidden-layer neural network). Consider a neural network $f : \mathbb{R}^d \to \mathbb{R}^K$ with input $\boldsymbol{x_0} \in \mathbb{R}^d$, a hidden layer with $U$ hidden neurons, and rectified linear unit (ReLU) activation function. If we sample uniformly in a ball $B_p(\boldsymbol{x_0}, R)$, then the cumulative distribution function of $\|\nabla g(\boldsymbol{x})\|_q$, denoted as $F_Y(y)$, is piece-wise linear with at most $M = \sum_{i=0}^{d} \binom{U}{i}$ pieces, where $g(\boldsymbol{x}) = f_c(\boldsymbol{x}) - f_j(\boldsymbol{x})$ for some given $c$ and $j$, and $\frac{1}{p} + \frac{1}{q} = 1, 1 \le p, q \le \infty$.

*Proof.* The $j_{\text{th}}$ output of a one-hidden-layer neural network can be written as

$$f_j(\boldsymbol{x}) = \sum_{r=1}^{U} \boldsymbol{V}_{jr} \cdot \sigma \left( \sum_{i=1}^{d} \boldsymbol{W}_{ri} \cdot x_i + b_r \right) = \sum_{r=1}^{U} \boldsymbol{V}_{jr} \cdot \sigma \left( \boldsymbol{w}_r \boldsymbol{x} + b_r \right),$$

where $\sigma(z) = \max(z, 0)$ is ReLU activation function, $\boldsymbol{W}$ and $\boldsymbol{V}$ are the weight matrices of the first and second layer respectively, and $\boldsymbol{w}_r$ is the $r_{\text{th}}$ row of $\boldsymbol{W}$. Thus, we can compute $g(\boldsymbol{x})$ and $\|\nabla g(\boldsymbol{x})\|_q$ below:

$$g(\boldsymbol{x}) = f_c(\boldsymbol{x}) - f_j(\boldsymbol{x}) = \sum_{r=1}^{U} \boldsymbol{V}_{cr} \cdot \sigma \left( \boldsymbol{w}_r \boldsymbol{x} + b_r \right) - \sum_{r=1}^{U} \boldsymbol{V}_{jr} \cdot \sigma \left( \boldsymbol{w}_r \boldsymbol{x} + b_r \right)$$

$$= \sum_{r=1}^{U} (\boldsymbol{V}_{cr} - \boldsymbol{V}_{jr}) \cdot \sigma \left( \boldsymbol{w}_r \boldsymbol{x} + b_r \right)$$

and

$$\|\nabla g(\boldsymbol{x})\|_q = \left\| \sum_{r=1}^{U} \mathbb{I}(\boldsymbol{w}_r \boldsymbol{x} + b_r)(\boldsymbol{V}_{cr} - \boldsymbol{V}_{jr}) \boldsymbol{w}_r^\top \right\|_q,$$

where $\mathbb{I}(z)$ is an univariate indicator function:

$$\mathbb{I}(z) = \left\{ \begin{array}{ll} 1, & \text{if } z > 0, \\ 0, & \text{if } z \le 0. \end{array} \right.$$

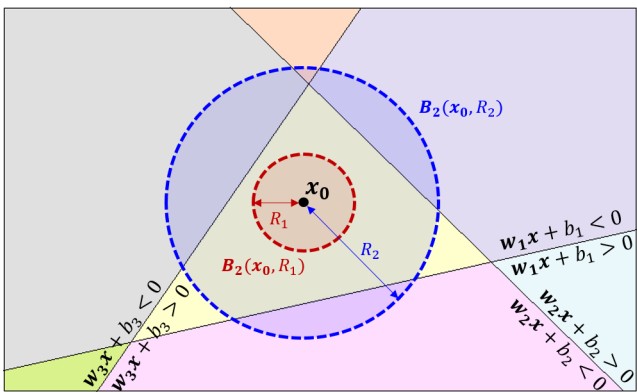

Figure 8: Illustration of Theorem D.1 with $d = 2, q = 2$ and $U = 3$. The three hyperplanes $\boldsymbol{w}_i \boldsymbol{x} + b_i = 0$ divide the space into seven regions (with different colors). The red dash line encloses the ball $B_2(\boldsymbol{x_0}, R_1)$ and the blue dash line encloses a larger ball $B_2(\boldsymbol{x_0}, R_2)$. If we draw samples uniformly within the balls, the probability of $\|\nabla g(\boldsymbol{x})\|_2 = y$ is proportional to the intersected volumes of the ball and the regions with $\|\nabla g(\boldsymbol{x})\|_2 = y$.

As illustrated in Figure 8, the hyperplanes $\boldsymbol{w}_r \boldsymbol{x} + b_r = 0, r \in \{1, \ldots, U\}$ divide the $d$ dimensional spaces $\mathbb{R}^d$ into different regions, with the interior of each region satisfying a different set of inequality constraints, e.g. $\boldsymbol{w}_{r_+} \boldsymbol{x} + b_{r_+} > 0$ and $\boldsymbol{w}_{r_-} \boldsymbol{x} + b_{r_-} < 0$. Given $\boldsymbol{x}$, we can identify which region it belongs to by checking the sign of $\boldsymbol{w}_r \boldsymbol{x} + b_r$ for each $r$. Notice that the gradient norm is the same for all the points in the same region, i.e. for any $\boldsymbol{x}_1, \boldsymbol{x}_2$ satisfying $\mathbb{I}(\boldsymbol{w}_r \boldsymbol{x}_1 + b_r) = \mathbb{I}(\boldsymbol{w}_r \boldsymbol{x}_2 + b_r) \, \forall r$,

we have $\|\nabla g(\boldsymbol{x}_1)\|_q = \|\nabla g(\boldsymbol{x}_2)\|_q$. Since there can be at most $M = \sum_{i=0}^{d} \binom{U}{i}$ different regions for a $d$-dimensional space with $U$ hyperplanes, $\|\nabla g(\boldsymbol{x})\|_q$ can take at most $M$ different values.

Therefore, if we perform uniform sampling in a ball $B_p(\boldsymbol{x_0}, R)$ centered at $\boldsymbol{x_0}$ with radius $R$ and denote $\|\nabla g(\boldsymbol{x})\|_q$ as a random variable $Y$, the probability distribution of $Y$ is discrete and its CDF is piece-wise constant with at most $M$ pieces. Without loss of generality, assume there are $M_0 \le M$ distinct values for $Y$ and denote them as $m_{(1)}, m_{(2)}, \ldots, m_{(M_0)}$ in an increasing order, the CDF of $Y$, denoted as $F_Y(y)$, is the following:

$$F_Y(m_{(i)}) = F_Y(m_{(i-1)}) + \frac{\mathbb{V}_d(\{\boldsymbol{x} \mid \|\nabla g(\boldsymbol{x})\|_q = m_{(i)}\}) \cap \mathbb{V}_d(B_p(\boldsymbol{x_0}, R)))}{\mathbb{V}_d(B_p(\boldsymbol{x_0}, R))}, i = 1, \ldots, M_0,$$

where $F_Y(m_{(0)}) = 0$ with $m_{(0)} < m_{(1)}$, $\mathbb{V}_d(E)$ is the volume of $E$ in a $d$ dimensional space. $\qquad \square$

## E    ADDITIONAL EXPERIMENTAL RESULTS

### E.1    PERCENTAGE OF EXAMPLES HAVING P VALUE $> 0.05$

Table 5 shows the percentage of examples where the null hypothesis cannot be rejected by K-S test, indicating that the maximum gradient norm samples fit reverse Weibull distribution well.

Table 5: Percentage of estimations where the null hypothesis cannot be rejected by K-S test for a significance level of 0.05. The bar plots of this table are illustrated in Figure 3.

|  | Least Likely | | Random | | Top-2 | |
|---|---|---|---|---|---|---|
|  | $L_2$ | $L_\infty$ | $L_2$ | $L_\infty$ | $L_2$ | $L_\infty$ |
| MNIST-MLP | 100.0 | 100.0 | 100.0 | 100.0 | 100.0 | 100.0 |
| MNIST-CNN | 99.6 | 99.8 | 99.2 | 100.0 | 99.4 | 100.0 |
| MNIST-DD | 99.8 | 100.0 | 99.6 | 99.8 | 99.8 | 99.8 |
| MNIST-BReLU | 93.3 | 95.4 | 96.8 | 96.8 | 97.6 | 98.2 |
| CIFAR-MLP | 100.0 | 100.0 | 100.0 | 100.0 | 100.0 | 100.0 |
| CIFAR-CNN | 100.0 | 100.0 | 100.0 | 100.0 | 100.0 | 100.0 |
| CIFAR-DD | 99.7 | 99.5 | 100.0 | 100.0 | 99.7 | 99.7 |
| CIFAR-BReLU | 99.5 | 99.2 | 100.0 | 100.0 | 99.7 | 99.7 |
| Inception-v3 | 100.0 | 100.0 | 100.0 | 100.0 | 100.0 | 100.0 |
| Resnet-50 | 99.0 | 100.0 | 100.0 | 100.0 | 100.0 | 100.0 |
| MobileNet | 100.0 | 100.0 | 100.0 | 100.0 | 98.0 | 99.0 |

### E.2    CLEVER V.S. NUMBER OF SAMPLES

Figure 9 shows the $\ell_2$ CLEVER score with different number of samples ($N_b = 50, 100, 250, 500$) for MNIST and CIFAR models. For most models except MNIST-BReLU, reducing the number of samples only change CLEVER scores very slightly. For MNIST-BReLU, increasing the number of samples improves the estimated lower bound, suggesting that a larger number of samples is preferred. In practice, we can start with a relatively small $N_b = a$, and also try $2a, 4a, \cdots$ samples to see if CLEVER scores change significantly. If CLEVER scores stay roughly the same despite increasing $N_b$, we can conclude that using $N_b = a$ is sufficient.

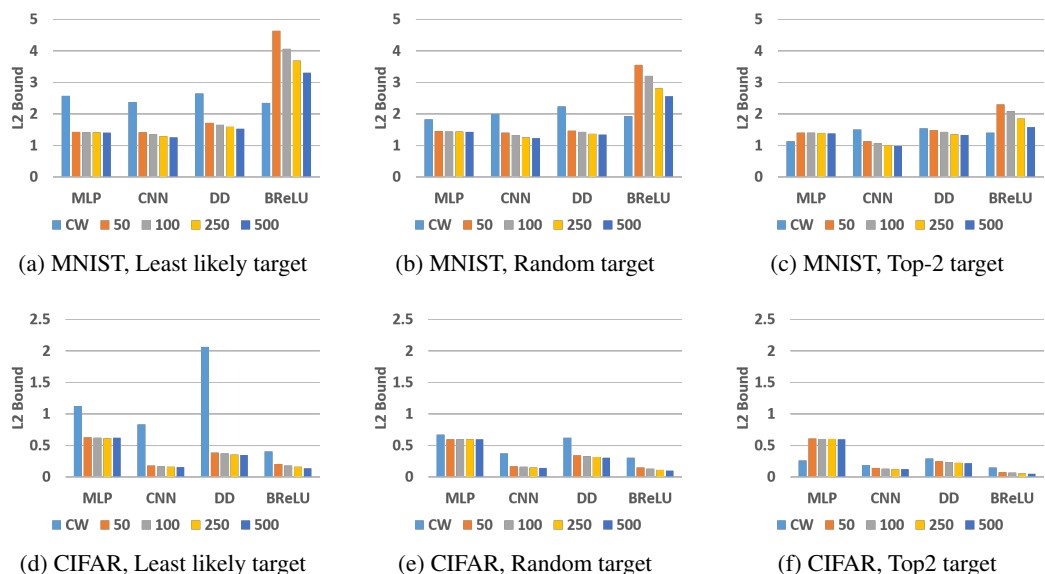

Figure 9: Comparison of the CLEVER score calculated by $N_b = \{50, 100, 250, 500\}$ and the $\ell_2$ norm of adversarial distortion found by CW attack (CW) on MNIST and CIFAR models with 3 target types.

