# OpenReview forum: "Evaluating the Robustness of Neural Networks: An Extreme Value Theory Approach"
_ICLR.cc/2018/Conference — Accept (Poster)_

### Official Review · AnonReviewer2 · 2017-11-10
**The authors present an "attack-independent" method for evaluating the robustness of a network. The idea is interesting, but the theoretical is very weak.**

**Rating:** 7
**Confidence:** 3

**Review:**

Summary
========

The authors present CLEVER, an algorithm which consists in evaluating the (local) Lipschitz constant of a trained network around a data point. This is used to compute a lower-bound on the minimal perturbation of the data point needed to fool the network.

The method proposed in the paper already exists for classical function, they only transpose it to neural networks. Moreover, the lower bound comes from basic results in the analysis of Lipschitz continuous functions.


Clarity
=====

The paper is clear and well-written.


Originality
=========

This idea is not new: if we search for "Lipschitz constant estimation" in google scholar, we get for example
Wood, G. R., and B. P. Zhang. "Estimation of the Lipschitz constant of a function." (1996)
which presents a similar algorithm (i.e., estimation of the maximum slope with reverse Weibull).


Technical quality
==============

The main theoretical result in the paper is the analysis of the lower-bound on \delta, the smallest perturbation to apply on
a data point to fool the network. This result is obtained almost directly by writing the bound on Lipschitz-continuous function
 | f(y)-f(x) | < L || y-x ||
where x = x_0 and y = x_0 + \delta.

Comments:
- Lemma 3.1: why citing Paulavicius and Zilinskas for the definition of Lipschitz continuity? Moreover, a Lipschitz-continuous function does not need to be differentiable at all (e.g. |x| is Lipschitz with constant 1 but sharp at x=0). Indeed, this constant can be easier obtained if the gradient exists, but this is not a requirement.

- (Flaw?) Theorem 3.2 : This theorem works for fixed target-class since g = f_c - f_j for fixed g. However, once g = min_j f_c - f_j, this theorem is not clear with the constant Lq. Indeed, the function g should be
g(x) = min_{k \neq c} f_c(x) - f_k(x).
Thus its Lipschitz constant is different, potentially equal to
L_q = max_{k} \| L_q^k \|,
where L_q^k is the Lipschitz constant of f_c-f_k. If the theorem remains unchanged after this modification, you should clarify the proof. Otherwise, the theorem will work with the maximum over all Lipschitz constants but the theoretical result will be weakened.

- Theorem 4.1: I do not see the purpose of this result in this paper. This should be better motivated.


Numerical experiments
====================

Globally, the numerical experiments are in favor of the presented method. The authors should also add information about the time it takes to compute the bound, the evolution of the bound in function of the number of samples and the distribution of the relative gap between the lower-bound and the best adversarial example.

Moreover, the numerical experiments look to be realized in the context of targeted attack. To show the real effectiveness of the approach, the authors should also show the effectiveness of the lower-bound in the context of non-targeted attack.


#######################################################

Post-rebuttal review
---------------------------

Given the details the authors provided to my review, I decided to adjust my score. The method is simple and shows to be extremely effective/accurate in practice.

Detailed answers:

1) Indeed, I was not aware that the paper only focuses on one dimensional functions. However, they still work with less assumption, i.e., with no differential functions. I was pointing out the similarities between their approach and your: the two algorithms (CLEVER and Slope) are basically the same, and using a limit you can go from "slope" to "gradient norm".
In any case, I have read the revision and the additional numerical experiment to compare Clever with their method is a good point.

2) " Overall, our analysis is simple and more intuitive, and we further facilitate numerical calculation of the bound by applying the extreme value theory in this work. "
This is right. I am just surprised is has not been done before, since it requires only few lines of derivation. I searched a bit but it is not possible to find any kind of similar results. Moreover, this leads to good performances, so there is no needs to have something more complex.

3) "The usual Lipschitz continuity is defined in terms of L2 norm and the extension to an arbitrary Lp norm is not straightforward"
Indeed, people usually use the Lipschitz continuity using the L2norm, but the original definition is wider.
Quickly, if you have a differential, scalar function from a space E -> R, then the gradient is a function from space E to E*, the dual of the space E.
Let || . || the norm of space E. Then, || . ||* is the dual norm of ||.||, and also the norm of E*.
In that case, Lipschitz continuity writes
f(x)-f(y) <= L || x-y ||, with L >= max_{x in E*} || f'(x) ||*
In the case where || . || is an \ell-p norm, then || . ||* is an \ell-q norm; with 1/p+1/q = 1.

If you are interested, there is a clear and concise explanation in the introduction of this paper: Accelerating the cubic regularization of Newton’s method on convex problems, by Yurii Nesterov.

I have no additional remarks for 4) -> 9), since everything is fixed in the new version of the paper.

---

> ### Author Response · Authors · 2017-12-19
> **Response to AnonReviewer 2 (part 3/3)**
>
>
> 7. Regarding the comment: “Globally, the numerical experiments are in favor of the presented method. The authors should also add information about the time it takes to compute the bound, the evolution of the bound in function of the number of samples and the distribution of the relative gap between the lower-bound and the best adversarial example.”:
>
> We thank the reviewer for this suggestion. Following your suggestion, we have included additional experimental results in Section 5.4 - Time v.s. Estimation Accuracy. In Figure 7, we vary the number of samples (N_b=50,100,250,500) and compute the L2 CLEVER scores for three large ImageNet models, Inception-v3, ResNet-50 and MobileNet. We observe that 50 or 100 samples are usually sufficient to get a reasonably accurate robustness estimation despite using a smaller number of samples. On a single GTX 1080 Ti GPU, the cost of 1 sample (with N_s = 1024 in Algorithm 1) is measured as 1.2 s for MobileNet, 5.5 s for ResNet-50 and 7.3 s for Inception-v3, thus the computational cost of CLEVER is feasible for state-of-the-art large-scale deep neural networks. Additional figures for MNIST and CIFAR datasets are given in Figure 9 in Appendix E2. We also added Figure 5 to show the empirical CDF of the gap between CLEVER score and the L2 distortion founded by CW attacks (the best attack) for 3 imagenet networks with random targets. It shows that at least 80% of the images have small gaps, demonstrating the effectiveness of our approach.
>
> 8. Regarding the comment: “Moreover, the numerical experiments look to be realized in the context of targeted attack. To show the real effectiveness of the approach, the authors should also show the effectiveness of the lower-bound in the context of non-targeted attack.”:
>
> We thank the reviewer for this important suggestion. Following your suggestion, we have added the experiments of un-targeted attack in Section 5.3. The results comparing average untargeted clever score and distortion found by CW and I-FGSM attacks are summarized in Table 2. We show that CW and I-FGSM attack results agree with the predicted robustness by CLEVER score, demonstrating the effectiveness of our approach.
>
> 9. Finally, we thank again the reviewer for the positive comments on the clarity of our paper and we hope our answers above were able to address all the comments regarding originality and technical contributions of our paper.  As suggested by the reviewer, in the current version of paper, we have included three sets of new experimental results regarding
> (1) untargeted attacks (Section 5.3, Table 2)
> (2) comparison to the slope sampling method of Wood & Zheng (1996) paper (Section 5.3, Table 3, Figure 4)
> (3) more numerical results of previous experiments (Section 5.3, Figure 5, Figure 7 and Figure 9)
> to show the advantage of our proposed method.
>
> As we highly value all reviewers’ inputs, we would like to use this opportunity to ask for your comments on the updated version during the author rebuttal stage. We believe we have carefully addressed all of your concerns, and we sincerely hope you could reconsider your decision.

---

> ### Author Response · Authors · 2017-12-19
> **Response to AnonReviewer 2 (part 2/3)**
>
>
> 4. Regarding the comment: “Moreover, a Lipschitz-continuous function does not need to be differentiable at all (e.g. |x| is Lipschitz with constant 1 but sharp at x=0). Indeed, this constant can be easier obtained if the gradient exists, but this is not a requirement”:
>
> We thank the reviewer for this comment. Indeed, as we show in Lemma 3.3, we can easily extend our analysis using the Lipschitz assumption to obtain the robustness guarantee for non-differentiable functions with a finite number of non-differentiable points (like networks with ReLU activations).
>
> 5. Regarding the comment: “(Flaw?) Theorem 3.2 : This theorem works for fixed target-class since g = f_c - f_j for fixed g. However, once g = min_j f_c - f_j, this theorem is not clear with the constant Lq. Indeed, the function g should be
> g(x) = min_{k \neq c} f_c(x) - f_k(x).
> Thus its Lipschitz constant is different, potentially equal to
> L_q = max_{k} \| L_q^k \|,
> where L_q^k is the Lipschitz constant of f_c-f_k. If the theorem remains unchanged after this modification, you should clarify the proof. Otherwise, the theorem will work with the maximum over all Lipschitz constants but the theoretical result will be weakened.”:
>
> We thank the reviewer for pointing out this potential ambiguity. There was an abuse of notation in Theorem 3.2 where the Lipschitz constant L_q is the lipschitz constant for function f_c-f_j, which is dependent on the index j. We have revised the notation accordingly in the revised paper and we use L_q^j to denote it is a Lipschitz constant of function (f_c- f_j) and is dependent on index j. For the untargeted attack that the reviewer is referring to, we note that Theorem 3.2 is indeed for un-targeted attacks, as it takes the min over all the targeted attack bound. We have made it clearer in the revised paper by adding a note of “Formal guarantee on lower bound for untargeted attack” in Theorem 3.2. In comparison, we also added Corollary 3.2.2 to give the formal guarantee for *targeted* attack. The algorithms for computing CLEVER for targeted and untargeted attacks are summarized in Algorithm 1 and 2 in Section 4.2. We note that we also included additional experiments for untargeted attacks in Table 2 in Section 5.3.
>
> 6. Regarding the comment: “ Theorem 4.1: I do not see the purpose of this result in this paper. This should be better motivated.”:
>
> We thank the reviewer for pointing this important observation. In the revised paper, we give a clearer explanation in the beginning of Section 4.1 of why we derive the CDF of $||\nabla g(x)||_q$. The reason is that in this work, we propose to use a new sampling method and extreme value theory to estimate the local Lipschitz constant; extreme value theory requires samples in a distribution of $||\nabla g(x)||_q$. A reader may wonder how the this distribution looks like. As an example, we show that we can derive the CDF of $||\nabla g(x)||_q$ for a 2-layer neural network with ReLU activation in Theorem 6.1 in Appendix D.

---

> ### Author Response · Authors · 2017-12-19
> **Response to AnonReviewer 2 (part 1/3)**
>
>
> We thank the reviewer for the positive comments on the clarity of our paper. However, we believe there might be some misunderstanding on the originality and technical quality of our work. Please allow us to clarify below.
>
> 1. Regarding the comment: “This idea is not new: if we search for "Lipschitz constant estimation" in google scholar, we get for example Wood, G. R., and B. P. Zhang. "Estimation of the Lipschitz constant of a function." (1996)  which presents a similar algorithm (i.e., estimation of the maximum slope with reverse Weibull)”:
>
> We thank the reviewer for pointing out this very early work of local Lipschitz constant estimation. We note that their sampling methodology is entirely different from our approach, as they estimate the Lipschitz constant by calculating the “slope” between pairs of sample points whereas in this paper we take the samples on the norm of the gradient directly. As “slope” is an approximation of gradient norm, it is conceivably (and also verified by our experiments in section 5.3, Table 3 and Figure 4) that the estimation will be less accurate than our method of directly computing the max norm of gradient. In addition, they only justified Lipschitz constant estimation for an *one-dimensional* function whereas our classifier function is very high-dimensional (d = 784 for MNIST, 3072 for CIFAR, 150,528 for ImageNet). In fact, how to accurately estimate Lipschitz constant for a high-dimensional function is still an open question. In this paper, we proposed to estimate Lipschitz constant by directly computing max norm of the gradient for the samples and using extreme value theory. As we show in Table 3 and Figure 4 in p.10 of the revised paper, Wood and Zhang’s (1996) approach (denoted as SLOPE) performs poorly on estimating Lipschitz constant for high-dimensional functions (i.e., neural net classifiers) and hence it is not suitable to use their method to evaluate adversarial perturbations in neural networks.
>
> 2. Regarding the comment: “The main theoretical result in the paper is the analysis of the lower-bound on \delta, the smallest perturbation to apply on a data point to fool the network. This result is obtained almost directly by writing the bound on Lipschitz-continuous function”:
>
> We thank the reviewer for this comment. Although our analysis is intuitive and straightforward, to the best of our knowledge, this is the *first* work that directly uses Lipschitz continuity to prove such a perturbation analysis. In comparison, Hein & Andriushchenko (2017) implicitly assumed Lipschitz continuity but used mean value theorem and Holder’s inequality in their analysis, which is not straightforward to achieve the same result, as also suggested by the reviewer. In addition to the difference in derivation of the bound, we would like to emphasize that our analysis can be easily extended to non-differentiable functions with a finite number of non-differentiable points, whereas Hein & Andriushchenko’s analysis is restricted to continuously differentiable functions. Overall, our analysis is simple and more intuitive, and we further facilitate numerical calculation of the bound by applying the extreme value theory in this work.
>
> 3. Regarding the comment: “Lemma 3.1: why citing Paulavicius and Zilinskas for the definition of Lipschitz continuity?”:
>
> Lemma 3.1 is not just the definition of Lipschitz continuity; it also gives the relationship between (local) Lipschitz constant in general Lp (p>=1)  norm and the dual norm of gradient. The usual Lipschitz continuity is defined in terms of L2 norm and the extension to an arbitrary Lp norm is not straightforward, thus we refer readers to Paulavicius and Zilinskas paper.

---

> ### Author Response · Authors · 2018-01-01
> **Please kindly provide inputs on our revised paper**
>
> Dear AnonReviewer 2,
>
> Following your valuable comments and suggestions, we have addressed the confusions in our theory, made comparisons with the additional reference you mentioned and added new numerical results. We have listed all changes we made in the general response to help you quickly find out the added materials. Thanks to your insightful comments, we believe our paper has been greatly improved after addressing all the concerns raised. For responses to any particular questions, please kindly read the corresponding section of our rebuttal.
>
> We will greatly appreciate it if you can provide new comments on the revised version of our paper. Thank you!
>
> Sincerely,
> Authors of Paper 767

---

### Official Review · AnonReviewer1 · 2017-11-27
**Critical problem and important claims, with experimental justification.**

**Rating:** 7
**Confidence:** 3

**Review:**

The work claims a measure of robustness of networks that is attack-agnostic. Robustness measure is turned into the problem of finding a local Lipschitz constant which is given by the maximum of the norm of the gradient of the associated function. That quantity is then estimated by sampling from the domain of maximization and observing the maximum value of the norm out of those samples. Such a maximum process is then described by the reverse Weibull distribution which is used in the estimation.

The paper closely follows Hein and Andriushchenko (2017). There is a slight modification that enlarges the class of functions for which the theory is applicable (Lemma 3.3). As far as I know, the contribution of the work starts in Section 4 where the authors show how to practically estimate the maximum process through back-prop where mini-batching helps increase the number of samples. This is a rather simple idea that is shown to be effective in Figure 3. The following section (the part starting from 5.3) presents the key to the success of the proposed measure.

This is an important problem and the paper attempts to tackle it in a computationally efficient way. The fact that the norms of attacks are slightly above the proposed score is promising, however, there is always the risk of finding a lower bound that is too small (zeros and large gaps in Figure 3). It would be nice to be able to show that one can find corresponding attacks that are not too far away from the proposed score.

Finally, a minor point: Definition 3.1 has a confusing notation, f is a K-valued vector throughout the paper but it also denotes the number that represents the prediction in Definition 3.1. I believe this is just a typo.

Edit: Thanks for the fixes and clarification of essential parts in the paper.

---

> ### Author Response · Authors · 2017-12-19
> **Response to AnonReviewer 1**
>
>
> 1. Regarding the comment: “The fact that the norms of attacks are slightly above the proposed score is promising, however, there is always the risk of finding a lower bound that is too small (zeros and large gaps in Figure 3). It would be nice to be able to show that one can find corresponding attacks that are not too far away from the proposed score”:
>
> We thank the reviewer for bringing this issue to our attention. Indeed, zero and small lower bounds were caused by the unstable MLE solver in scipy. We have fixed this issue by renormalizing samples before MLE and updated the results in Table 4 and Figure 6 in p.11 of the revised paper. In Figure 5, we show the empirical CDF of the gaps for 100 ImageNet images, and find that most gaps are indeed small. We also report the percentage of images where p-value in K-S test is greater than 0.05 in Figure 3 (p.8) and Table 5 (p.16). The numbers are all close to 100%, justifying the hypothesis that the sampled maximum gradient norms follow the reverse Weibull distribution.
>
> 2. Regarding the comment: “Finally, a minor point: Definition 3.1 has a confusing notation, f is a K-valued vector throughout the paper but it also denotes the number that represents the prediction in Definition 3.1. I believe this is just a typo”:
>
> We thank the reviewer for pointing out this typo. We have fixed the typos in Definition 3.1 accordingly.

---

### Official Review · AnonReviewer3 · 2017-12-01
**Interesting point of view on analysis of robustness**

**Rating:** 7
**Confidence:** 1

**Review:**

In this work, the objective is to analyze the robustness of a neural network to any sort of attack.

This is measured by naturally linking the robustness of the network to the local Lipschitz properties of the network function. This approach is quite standard in learning theory, I am not aware of how original this point of view is within the deep learning community.

This is estimated by obtaining values of the norm of the gradient (also naturally linked to the Lipschitz properties of the function) by backpropagation. This is again a natural idea.

---

> ### Author Response · Authors · 2017-12-19
> **Response to AnonReviewer 3**
>
>
> 1. Regarding the comment of using local Lipschitz properties of the network function:
>
> We thank the reviewer for pointing this out. We note that this paper is the *first* work to derive the lower bound of minimum distortion using (local) cross-Lipschitz continuity assumption. For continuously differentiable classification functions, we show that with the Lipschitz continuity assumption, our result is consistent with Hein & Andriushchenko (2017), who used Mean Value Theorem and Holder’s inequality to obtain the same lower bound. In addition, we show in Lemma 3.3 that our approach can easily extend to non-differentiable functions (e.g. ReLU activations), whereas the analysis in Hein & Andriushchenko (2017) is restricted to continuously differentiable functions.
>
> 2. Regarding the comment of using the norm of the gradient by backpropagation to estimate Lipschitz constant:
>
> We note that there exist other estimation methods, e.g. Wood & Zhang (1996) as mentioned by AnonReviewer 2, where they calculate the slope between pairs of sample points instead of taking the samples on the norm of the gradient in this paper. However, as shown in Table 3 and Figure 4 in p.10 of the revised paper, their approach (denoted as SLOPE) perform poorly on estimating Lipschitz constant for high-dimensional functions like neural networks, thus are not suitable to estimate minimum adversarial distortions.

---

### Author Response · Authors · 2017-12-19
**General Response to all the reviewers**

We thank AnonReviewer 1 and AnonReviewer 3 for the constructive comments and overall positive assessments. We also thank AnonReviewer 2 for the insightful comments and valuable suggestions. We will detail in our response below of how we have addressed these comments and reply to each reviewer in the comments. Besides, we believe there might be some misunderstanding on the originality and technical quality of our paper, which will also be clarified.

To improve the quality of this paper, we have added more theoretical results, figures, and experimental results to our revised version (uploaded). We summarize these changes as below:

* Summary of the changes:
Section 3:
1. We have made it clearer at the beginning of Section 3 that our robustness guarantee in this paper is more general and more intuitive than Hein & Andriushchenko (2017) by using Lipschitz continuity assumption. For continuously differentiable functions, our result is consistent with Hein & Andriushchenko (2017); our analysis can easily extend to non-differentiable functions (e.g. ReLU activations) whereas the analysis in Hein & Andriushchenko (2017) is restricted to continously differentiable functions.

2. We have added Corollary 3.2.2 as a formal guarantee of the targeted attack. We also added a note that Theorem 3.2 and Corollary 3.2.1 are formal guarantees for un-targeted attack.

3. We have changed the notation of Lipschitz constant of function (f_c-f_j) from L_q to L_q^j to make it clearer that it is dependent on the index j.

Section 4:
1. We have added a paragraph before Section 4.1 to comment on the difference between our approach and (Wood & Zhang, 1996) as mentioned by AnonReviewer 2. We note that the sampling methodology is entirely different and their approach (denoted as SLOPE) works poorly on estimating Lipschitz constant for high dimensional functions like neural networks, as demonstrated in our Table 3 and Figure 4 in p.10.

2. In Section 4.1, we gave a clearer explanation of why we want to derive the CDF of $||\nabla g(x)||_q$. The reason is that in this paper, we propose to sample the maximum of $||\nabla g(x)||_q$ to estimate local Lipschitz constant via extreme value theory. A reader may wonder how the distribution of $||\nabla g(x)||_q$ looks like. Thus, as an example, we show that we can derive the CDF of $||\nabla g(x)||_q$ for a 2-layer neural network with ReLU activation in Theorem 6.1 in Appendix D.

3. In Section 4.2, we added Algorithm 2 (clever-u) to illustrate how to compute the clever score for untargeted attacks. The original Algorithm 1 (clever-t) is for targeted attacks.

Section 5:
1. In Section 5.2, we reported the percentage of images where p-value in Kolmogorov-Smirnov test is greater than 0.05 in Figure 3 and Table 5 (in appendix E1). The numbers are all close to 100%, justifying the hypothesis that the sampled maximum gradient norms follow reverse Weibull distribution.

2. In Section 5.3, we added untargeted attack results in Table 2. We show that CW and I-FGSM attack results agree with the predicted robustness by CLEVER score.

3. In Section 5.3, we also implemented the method in Wood and Zhang (1996) to estimate Lipschitz constant and calculate the average L2 and L infinity distortion for targeted attacks in Table 3 (denoted as SLOPE) and Figure 4. We show that their method (SLOPE) gives poor estimates on the distortions for high dimensional functions like neural networks.

4. In Section 5.3, we also fixed an unstable MLE estimation issue in scipy by renormalizing samples before MLE and improved the results in Table 4 and Figure 6.

5. In Section 5.4, we reported the runtime for ImageNet networks - on a single GTX 1080 Ti GPU, the cost of 1 sample (with Ns = 1024 in Algorithm 1) is measured as 1.2 s for MobileNet, 5.5 s for ResNet-50 and 7.3 s for Inception-v3. Thus the computational cost of CLEVER is feasible for state-of-the-art large-scale deep neural networks. We also discussed how the number of samples affects estimation accuracy in Figure 7 for 3 imagenet models and Figure 9 for mnist and cifar in Appendix E2 - we observe that 50 or 100 samples are usually sufficient to get a reasonably accurate robustness estimation despite using a smaller number of samples. The results indicate that our method is practical for large networks.

---

### Public Comment · ~Ian_Goodfellow1 · 2018-04-30
**Rebuttal**

I've written a brief rebuttal of this paper here: https://arxiv.org/abs/1804.07870

In summary, an estimate of a lower bound is not a lower bound. The estimate used here can overestimate. Specifically, "gradient masking," one of the main factors causing attack-based methods to overestimate robustness, also causes CLEVER to overestimate.

---

> ### Author Response · Authors · 2018-05-01
> **Reply to “Gradient Masking causes CLEVER to overestimate adversarial perturbation size”**
>
>
> Hello Ian,
>
> Thanks for initiating the discussion. Based on your arxiv article, we believe there is some misunderstanding in your comments.
>
> Below we summarize our replies to your comments:
>
> 1. On overestimating perturbation size
>
> CLEVER intends to estimate the lower bound of minimum distortion, and we advocate the estimate as a robustness score. We would like to reiterate that the theoretical bound we provided in Section 3 of our paper is a certified lower bound, and CLEVER is a statistical estimate of it. Since it is a statistical estimate (i.e., an estimated lower bound), it is apparent that it is not certified. However, the empirical results in our paper show that in many neural networks CLEVER not only provides an attack-independent score that can quantify the robustness of a neural network classifier, but also serves as a valid lower bound. Moreover, CLEVER is the first score that can provide robustness estimate for large ImageNet networks and has inspired follow-up works in fast certified robustness verification, such as https://arxiv.org/abs/1804.09699
>
> 2. On gradient masking
>
> CLEVER is intended to be a tool for network designer and to evaluate network robustness in the “white-box” setting in which we know how a (defended) neural network processes the input. Therefore, the gradient masking “counterexample” made in the arxiv article, which takes the form g(h(x)) where g is an innate neural network and h is a function having zero gradient almost everywhere, can still adopt CLEVER for robustness evaluation by simply using the Backward Pass Differentiable Approximation (BPDA) method proposed by Athalye, Carlini and Wagner in https://arxiv.org/pdf/1802.00420.pdf, which is known to resolve the gradient obfuscation issues. In other words, when computing the gradient of g(h(x)) for CLEVER in your example, one should use BPDA at h(x) instead of directly computing gradient of g(h(x)) at x. Therefore,  your concerns can be easily addressed.
>
> 3. On numerical error on digital computers
>
> Sec 4 and Sec 5 in the arxiv article argue that on digital computers all functions will not be Lipschitz continuous and behave like the staircase function in Sec 3, where the gradient is zero almost everywhere so that CLEVER will not work. This conclusion is incorrect. Because on digital computers and under the white-box setting, gradients can be computed via automatic differentiation, which takes care of the numerical issues you mentioned and is well supported by mature packages like Tensorflow.
>
> We plan to explicitly include BPDA in CLEVER computation. Please stay tuned to our GitHub repo and IBM ART toolbox for updates:
> https://github.com/huanzhang12/CLEVER/
> https://github.com/IBM/adversarial-robustness-toolbox
>
>
> Best,
>
> CLEVER Authors

---

### Decision · Program_Chairs · 2018-01-29
**ICLR 2018 Conference Acceptance Decision**

**Decision:**

Accept (Poster)

**Comment:**

This paper proposes a new metric to evaluate the robustness of neural networks to adversarial attacks. This metric comes with theoretical guarantees and can be efficiently computed on large-scale neural networks.

Reviewers were generally positive about the strengths of the paper, especially after major revisions during the rebuttal process. The AC believes this paper will contribute to the growing body of literature in robust training of neural networks.